# Novel tau biomarkers phosphorylated at T181, T217, or T231 rise in the initial stages of the preclinical Alzheimer's *continuum* when only subtle changes in Aβ pathology are detected

Marc Suárez-Calvet[1,2,3,4,†] (ID), Thomas K Karikari[5,†], Nicholas J Ashton[5,6,7,8] (ID), Juan Lantero Rodríguez[5], Marta Milà-Alomà[1,2,4,9], Juan Domingo Gispert[1,2,9,10], Gemma Salvadó[1,2], Carolina Minguillon[1,2,4], Karine Fauria[1,4], Mahnaz Shekari[1,2,9], Oriol Grau-Rivera[1,2,3,4], Eider M Arenaza-Urquijo[1,2,4], Aleix Sala-Vila[1,2], Gonzalo Sánchez-Benavides[1,2,4], José Maria González-de-Echávarri[1,2], Gwendlyn Kollmorgen[11], Erik Stoops[12], Eugeen Vanmechelen[12], Henrik Zetterberg[5,13,14,15], Kaj Blennow[5,13,*] (ID), José Luis Molinuevo[1,2,4,9,**] (ID) & for the ALFA Study[‡]

## Abstract

In Alzheimer's disease (AD), tau phosphorylation in the brain and its subsequent release into cerebrospinal fluid (CSF) and blood is a dynamic process that changes during disease evolution. The main aim of our study was to characterize the pattern of changes in phosphorylated tau (p-tau) in the preclinical stage of the Alzheimer's *continuum*. We measured three novel CSF p-tau biomarkers, phosphorylated at threonine-181 and threonine-217 with an N-terminal partner antibody and at threonine-231 with a mid-region partner antibody. These were compared with an automated mid-region p-tau181 assay (Elecsys) as the gold standard p-tau measure. We demonstrate that these novel p-tau biomarkers increase more prominently in preclinical Alzheimer, when only subtle changes of amyloid-β (Aβ) pathology are detected, and can accurately differentiate Aβ-positive from Aβ-negative cognitively unimpaired individuals. Moreover, we show that the novel plasma N-terminal p-tau181 biomarker is mildly but significantly increased in the preclinical stage. Our results support the idea that early changes in neuronal tau metabolism in preclinical Alzheimer, likely in response to Aβ exposure, can be detected with these novel p-tau assays.

**Keywords** Alzheimer's disease; biomarker; cerebrospinal fluid; plasma; tau
**Subject Categories** Biomarkers; Neuroscience

## Introduction

Cerebrospinal fluid (CSF) phosphorylated tau (p-tau) is, together with CSF amyloid-β 42 (Aβ42) and CSF total tau (t-tau), a core

1  Barcelonaβeta Brain Research Center (BBRC), Pasqual Maragall Foundation, Barcelona, Spain
2  IMIM (Hospital del Mar Medical Research Institute), Barcelona, Spain
3  Servei de Neurologia, Hospital del Mar, Barcelona, Spain
4  Centro de Investigación Biomédica en Red de Fragilidad y Envejecimiento Saludable (CIBERFES), Madrid, Spain
5  Department of Psychiatry and Neurochemistry, Institute of Neuroscience and Physiology, The Sahlgrenska Academy, University of Gothenburg, Gothenburg, Sweden
6  Department of Psychiatry and Neurochemistry, Institute of Neuroscience and Physiology, Wallenberg Centre for Molecular and Translational Medicine, The Sahlgrenska Academy at the University of Gothenburg, Gothenburg, Sweden
7  Institute of Psychiatry, Psychology & Neuroscience, King's College London, Maurice Wohl Clinical Neuroscience Institute, London, UK
8  NIHR Biomedical Research Centre for Mental Health & Biomedical Research Unit for Dementia at South London & Maudsley NHS Foundation, London, UK
9  Universitat Pompeu Fabra, Barcelona, Spain
10  Centro de Investigación Biomédica en Red Bioingeniería, Biomateriales y Nanomedicina, Madrid, Spain
11  Roche Diagnostics GmbH, Penzberg, Germany
12  ADx NeuroSciences, Ghent, Belgium
13  Clinical Neurochemistry Laboratory, Sahlgrenska University Hospital, Mölndal, Sweden
14  Department of Neurodegenerative Disease, UCL Institute of Neurology, London, UK
15  UK Dementia Research Institute at UCL, London, UK
   *Corresponding author. Tel: +46 0313421000; E-mail: kaj.blennow@neuro.gu.se
   **Corresponding author. Tel: +34 933160990; E-mail: jlmolinuevo@barcelonabeta.org
   †These authors contributed equally to this work
   ‡The complete list of collaborators of the ALFA Study can be found in the acknowledgements section.

biomarker for Alzheimer's disease (AD). Overwhelming evidence indicates that CSF p-tau is increased in patients with AD (both in the prodromal and dementia stages) compared to controls (Hansson et al, 2006; Shaw et al, 2009; Mattsson et al, 2009; Olsson et al, 2016). Moreover, CSF p-tau correlates with cognitive impairment better than Aβ-related biomarkers (Gómez-Isla et al, 1997; Blennow et al, 2010; Nelson et al, 2012; Roe et al, 2013; Jack et al, 2018; Aschenbrenner et al, 2018). CSF p-tau is useful to stage the disease, although longitudinal studies suggest that it may decrease in late stages of AD (Fagan et al, 2014; McDade et al, 2018; Sutphen et al, 2018; Lleó et al, 2019; Schindler et al, 2019). CSF p-tau is also an excellent prognostic biomarker in AD since it finely predicts progression from cognitively unimpaired (CU) to mild cognitive impairment (MCI) and, eventually, to AD dementia (Roe et al, 2013; Petersen et al, 2013; Ferreira et al, 2014). Furthermore, CSF p-tau is increased in preclinical AD (Hansson et al, 2006; Shaw et al, 2009; Bateman et al, 2012).

In AD, tau aggregated in the neurofibrillary tangles (NFTs) is aberrantly hyperphosphorylated and several phosphorylated sites have been identified (Grundke-Iqbal et al, 1986; Ksiezak-Reding et al, 1988; Goedert et al, 1988, 1989; Lee et al, 1991, 2001). Yet, in the CSF biomarkers field, the most common tau phosphorylation site used as a target is threonine-181 (p-tau181) (Blennow et al, 1995). In fact, it is usually assumed that the term "p-tau" refers to phosphorylation at threonine-181 if not otherwise specified. Other tau phosphorylation sites have been investigated in the CSF of AD patients including mid-region residues p199, p212/p214, p217, p231, p231/p235 and the C-terminal residues p396/p404 (Ishiguro et al, 1999; Kohnken et al, 2000; Hu et al, 2002; Buerger et al, 2002b; Hampel et al, 2004; Singer et al, 2009; Meredith et al, 2013; Russell et al, 2016; Janelidze et al, 2020b). A very recent study in the Dominantly Inherited Alzheimer Network (DIAN) cohort finely showed the pattern of changes of p-tau phosphorylation in the early stages of autosomal-dominant AD and also in a group of preclinical sporadic AD (Barthélemy et al, 2020c).

Similarly, it is also assumed that the term "p-tau" refers to the mid-region fragment, where most commercially available p-tau assays are targeted to. It is known, however, that CSF contains a mix of both N-terminal and mid-region fragments, while C-terminal fragments are considerably less abundant (Meredith et al, 2013; Barthélemy et al, 2016; Sato et al, 2018; Cicognola et al, 2019; Chen et al, 2019). Taking all this into account has been a key factor for the development of blood tests targeting p-tau. The use of antibodies targeting N-terminal fragments has allowed us and others to successfully measure p-tau181 in plasma, which accurately detects AD and discriminates it from other neurological diseases (Tatebe et al, 2017; Mielke et al, 2018; Karikari et al, 2020; Thijssen et al, 2020; Janelidze et al, 2020a). Moreover, plasma p-tau181 starts to increase in preclinical AD and further increases in the MCI and dementia stages (Karikari et al, 2020; Janelidze et al, 2020a). Besides p-tau181, recent strong data indicates that p-tau217, measured both in CSF and in plasma, accurately predicts Aβ pathology in both symptomatic and asymptomatic stages and is an excellent biomarker to discriminate AD from healthy controls and other neurodegenerative diseases (Barthelemy et al, 2015; preprint: Barthélemy et al, 2017; Barthélemy et al, 2020a, 2020b, 2020c; Janelidze et al, 2020b; Palmqvist et al, 2020). In autosomal-dominant AD, CSF p-tau217 even increases two decades before tau PET (Barthélemy et al, 2020c).

Despite the recent breakthrough developments in understanding p-tau as both a CSF and blood biomarker, it is less known whether specific p-tau biomarkers change early in the continuum of sporadic Alzheimer, when only subtle, incipient changes in Aβ pathology are present. Studying these early stages of the disease is particularly relevant for the p-tau biomarkers since their changes probably precede NFT pathology (Barthélemy et al, 2020c). Amid the new p-tau assays being developed (targeting different phosphorylation sites, targeting N-terminal vs. mid-region tau or using different platforms), it is also important to perform a head-to-head comparison of these assays. Moreover, there are promising tau phosphorylations, such p-tau231, that have not been yet investigated in asymptomatic stages. In this study, we aimed at characterizing the pattern of changes in phosphorylated tau in the preclinical stage of the Alzheimer's continuum. For this purpose, we measured a set of novel CSF p-tau biomarkers in the ALFA+ study (a cohort of CU individuals, some of whom are in the preclinical stage of the Alzheimer's continuum) and compared them with the well-established Elecsys® Mid-p-tau181 assay [targeting mid-region (Mid) tau fragments phosphorylated at threonine-181 (Lifke et al, 2019)], used herein as the reference assay. The set of novel p-tau biomarkers studied include (Fig EV1): (a) N-p-tau181 [targeting tau forms phosphorylated at threonine-181 and containing the N-terminal (N) epitope 6-18), (b) N-p-tau217 (targeting tau forms phosphorylated at threonine-217 as well as containing the N-terminal epitope 6–18) and (c) Mid-p-tau231 (targeting Mid tau fragments phosphorylated at threonine-231). Moreover, we investigated, using the N-p-tau181 assay in plasma, if preclinical disease changes can be reliably detected in blood and how this compared against the CSF biomarkers and also against plasma neurofilament light (NfL), the most widely used blood biomarker. We tested the hypothesis of whether these novel p-tau biomarkers change in initial stages in the disease process, where only subtle changes in Aβ are detectable. In order to ensure robustness, we used both CSF and PET as biomarkers of Aβ pathology, which reflect different aspects of Aβ pathology (i.e. soluble Aβ and fibrillar Aβ aggregates, respectively) (Dubois et al, 2016). Based on our findings, we propose a model of changes in p-tau biomarkers in the preclinical stage of the sporadic Alzheimer's continuum.

## Results

### Participants' characteristics and the effect of age and sex on p-tau biomarkers

The first consecutive 381 participants of the ALFA+ cohort were included in this study. Their demographic and clinical features as well as fluid biomarker concentrations are shown in Table 1. Participants were classified as Aβ-negative (A−; $n = 250$) or Aβ-positive (A+; $n = 131$) using a previously established cut-off value for the CSF Aβ42/40 ratio of 0.071 (Milà-Alomà et al, 2020). Participants in the A+ group were older, but there were no differences in education, global cognition [as measured by the Mini-Mental State Examination (MMSE)] and sex distribution. As expected, the A+ group had a higher percentage of APOE-ε4 carriers, Aβ PET-positive visual reads and a higher [18F]flutemetamol uptake in Aβ PET, as measured by the Centiloid (CL) scale. The AD CSF biomarkers CSF t-tau as well

**Table 1. Participants' characteristics and biomarkers by Aβ status group.**

| | Total (n = 381) | A− (n = 250, 65.6%) | A+ (n = 131, 34.4%) | P-value* |
|---|---|---|---|---|
| **Demographics, clinical characteristics** | | | | |
| Age, years | 61.2 (4.68) | 60.6 (4.44) | 62.3 (4.93) | 0.0007* |
| Female, n (%) | 232 (60.9) | 155 (62.0) | 77 (58.8) | 0.541 |
| Education, years | 13.4 (3.51) | 13.5 (3.48) | 13.2 (3.57) | 0.467 |
| APOE-ε4 carriers, n (%) | 201 (52.8) | 102 (40.8) | 99 (75.6) | <0.0001* |
| MMSE | 29.1 (0.954) | 29.1 (0.935) | 29.2 (0.993) | 0.608 |
| Centiloids[a] | 2.82 (16.8) | −4.54 (6.37) | 16.83 (21.1) | <0.0001* |
| Aβ PET-positive (VR), n (%)[a] | 42 (12.8) | 3 (1.4) | 39 (34.8) | <0.0001* |
| **p-tau-related biomarkers** | | | | |
| CSF Mid-p-tau181 (pg/ml) | 16.3 (7.69) | 14.5 (5.23) | 19.8 (10.1) | <0.0001* |
| CSF N-p-tau181 (pg/ml) | 346 (216) | 278 (97.7) | 477 (304) | <0.0001* |
| CSF N-p-tau217 (pg/ml) | 6.29 (6.41) | 4.07 (2.28) | 10.5 (9.07) | <0.0001* |
| CSF Mid-p-tau231 (pg/ml) | 8.29 (6.08) | 6.11 (2.28) | 12.5 (8.46) | <0.0001* |
| Plasma N-p-tau181 (pg/ml) | 9.55 (3.86) | 8.83 (3.21) | 10.9 (4.57) | <0.0001* |
| **Core AD biomarkers** | | | | |
| CSF Aβ42/40 | 0.075 (0.020) | 0.087 (0.010) | 0.051 (0.012) | <0.0001* |
| CSF t-tau (pg/ml) | 199 (74.1) | 182 (58.8) | 230 (90.0) | <0.0001* |
| CSF NfL (pg/ml) | 82.8 (30.6) | 78.1 (28.3) | 92.0 (32.7) | 0.0006* |
| Plasma NfL (pg/ml) | 10.4 (3.71) | 9.83 (3.27) | 11.6 (4.22) | 0.0006* |

Aβ42, amyloid-β 42; Aβ40, amyloid-β 40; CSF, cerebrospinal fluid; Mid, mid-region; MMSE, Mini-Mental State Examination; N, N-terminal; NfL, neurofilament; p-tau, phosphorylated tau; t-tau, total tau; PET, positron emission tomography; VR, visual read.
Participants were classified as Aβ-negative (A−) or Aβ-positive (A+) as defined by the mean Aβ42/40 ratio. Data are expressed as mean (M) and standard deviation (SD) or number of participants (n) and percentage (%), as appropriate. A t-test was used to compared age, education, MMSE and Centiloids between Aβ status groups and Pearson's chi-square to compare sex, APOE-ε4 and Aβ PET visual read positivity frequency between Aβ status groups. All biomarkers were compared with a one-way ANCOVA adjusted by age and sex. The P-values indicated in the last column refer to the Aβ status group effect.
[a]Aβ PET was available in 331 participants and, among them, two did not have visual read assessment.
*Significant differences.

as CSF and plasma NfL were also significantly increased in the A+ group (Table 1).

In the whole cohort, there was a significant increase in all CSF p-tau biomarkers ($P < 0.0001$) and plasma N-p-tau181 ($P = 0.001$) as a function of age (Fig 1A–E; Table EV1). When stratified by Aβ status, we observed that the increase in all the CSF p-tau biomarkers as a function of age only occurred in the A+ group but not in the A− one (Fig 1A–D; Table EV1), as also shown by the significant "Age × Aβ status" interaction terms, but not in plasma N-p-tau181.

Interestingly, after adjusting for age, levels of CSF N-p-tau217 were slightly, but significantly, higher in women (M = 6.71 pg/ml, SD = 7.01) than men (M = 5.63 pg/ml, SD = 5.31; $P = 0.016$). This difference remained significant after additionally adjusting for Aβ status ($P = 0.005$). The rest of the CSF or plasma p-tau biomarkers were not affected by sex.

**Association of p-tau biomarkers and Aβ pathology**

In order to investigate the relationship between Aβ pathology and the novel p-tau biomarkers, we first tested whether each of the p-tau biomarkers differed between the A+ and A− groups, as defined by the CSF Aβ42/40 ratio. We observed that all p-tau biomarkers were significantly higher in the A+ group, after adjusting for the

effect of age and sex (Table 1; Fig 2A–E). Specifically, the increase in the A+ group compared to the A− one was 36.6% for CSF Mid-p-tau181, 71.6% for CSF N-p-tau181, 158% for N-p-tau217 and 105% for Mid-p-tau231. Remarkably, the relative increase of these CSF p-tau biomarkers was considerably higher than those of CSF t-tau (26.4%) and CSF NfL (17.8%; Table 1). Moreover, plasma N-p-tau181 was 23.4% higher in the A+ group than in the A− one (Fig 2E; Table 1), a greater increase than that of plasma NfL (18.0%), the most used AD plasma biomarker to date.

Next, we assessed the associations between the p-tau biomarkers and CSF Aβ42/40 ratio as a continuous variable and observed that all CSF p-tau biomarkers, as well as plasma N-p-tau181, significantly increased ($P < 0.0001$) as a function of higher Aβ pathology (that is, lower CSF Aβ42/40 ratio). However, these associations were modified by the Aβ status for all p-tau biomarkers, as indicated by the significant "CSF Aβ42/40 × Aβ status" interaction term (Table 2, Fig 2F–J). After stratifying by Aβ status (cut-off for CSF Aβ42/40 is depicted with a green dashed line in Fig 2F–J), all p-tau biomarkers significantly increased while the CSF Aβ42/40 ratio decreased in the A+ group (Table 2, Fig 2F–J), a result that was not observed in the A− group.

We also assessed the relationship between the novel p-tau biomarkers and another Aβ biomarker, Aβ PET, which was available in 331 (86.9%) of the studied participants. We confirmed that

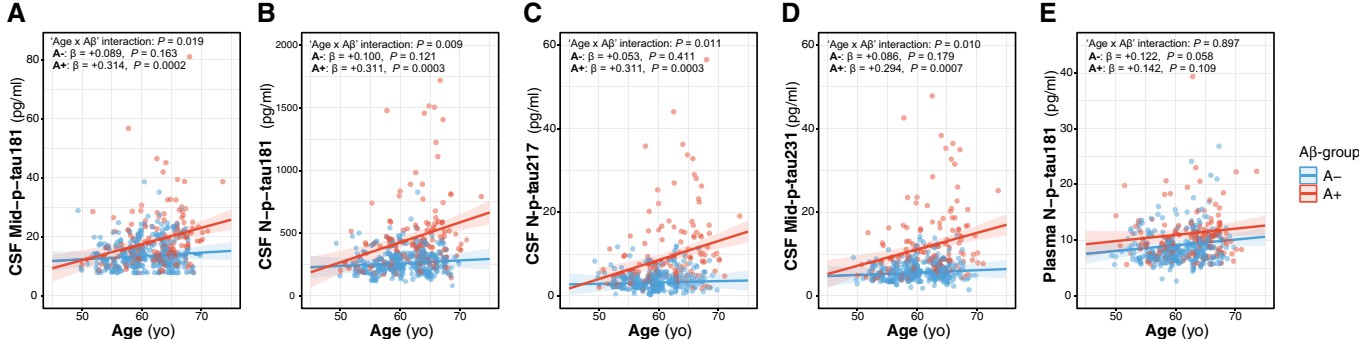

**Figure 1. P-tau biomarkers change with age.**

A–E   Scatter plots showing the association of each of the p-tau biomarkers with age in the Aβ-negative (A−; blue; n = 250) and the Aβ-positive (A+; red; n = 131) groups. The solid lines indicate the regression line and the 95% confidence intervals for each of the groups. For each group, the standardized regression coefficients (β) and the P-values were computed using a linear model adjusting for sex. Additionally, we computed the "Age × Aβ status" interaction term. Abbreviations: CSF, cerebrospinal fluid; Mid, mid-region; N, N-terminal; p-tau, phosphorylated tau.

all p-tau biomarkers were significantly higher in the A+ (as defined by the Aβ PET visual read) compared to the A− group (Fig 3A–E). In line with the CSF Aβ42/40 results, all p-tau biomarkers were positively associated with the CL scale, a standardized measure of Aβ PET uptake, in the whole sample (P < 0.0001 for all p-tau biomarkers; Fig 3F–J). When stratifying for A+ or A− groups (by Aβ PET visual read), we also observed that all p-tau biomarkers increased as a function of the CL scale (Table 2). Remarkably, and unlike the CSF Aβ42/40 ratio, CSF N-p-tau181, N-p-tau217 and Mid-p-tau231 and plasma N-p-tau181 biomarkers also increased as a function of the CL scale in the A− group, which did not occur with CSF Mid-p-tau181, suggesting an early increase of the novel p-tau biomarkers with subthreshold levels of Aβ PET (Table 2). We repeated the same analyses after defining the A+ group as having a CL > 12 in Aβ PET, instead of a positive visual read, and the results were similar (Table EV2).

**P-tau biomarkers and subthreshold Aβ PET**

It is well known that Aβ PET changes occur later than the CSF Aβ42/40 ratio in the Alzheimer's *continuum*, and we therefore hypothesized that the novel p-tau biomarkers studied herein start to increase very early in the *continuum* when Aβ PET is still negative. To test this hypothesis, we investigated the p-tau biomarkers in subthreshold levels of Aβ PET. For these purposes, we used two CL scale cut-offs: 12 CL and 30 CL (both depicted in green dashed lines in Fig 3F–J). The 12 CL cut-off was previously proposed to detect subtle Aβ pathology in a comparison against CSF Aβ42 and neuropathological assessment of Aβ plaques (La Joie *et al*, 2019; Salvadó *et al*, 2019). In contrast, 30 CL is a cut-off that is in the range of agreement with the visual read method in clinical populations (24–35 CL) and therefore reflects established Aβ pathology (Leuzy *et al*, 2016; Rowe *et al*, 2017; La Joie *et al*, 2019; Salvadó *et al*, 2019; Amadoru *et al*, 2020). We observed that all CSF and plasma p-tau biomarkers were increased in the group of participants with CL > 30 with respect to the negative group (CL ≤ 12; Fig 3K–O). Interestingly, however, during the intermediate period between 12 CL and 30 CL (when there is subtle Aβ pathology), CSF N-p-tau181, N-p-tau217 and Mid-p-tau231 (but not

CSF Mid-p-tau181) were already significantly higher than the ≤ 12 CL group (Fig 3K–O). Specifically, the increases in the 12–30 CL group compared to the CL ≤ 12 were the following: 44.1% for CSF N-p-tau181, 84.5% for CSF N-p-tau217 and 64.5% for Mid-p-tau231. Importantly, plasma N-p-tau181 had a significant 23.6% increase in the 12–30 CL group compared to the ≤ 12 CL group (Fig 3O).

Altogether, these results indicate that the novel CSF N-p-tau181, N-p-tau217 and Mid-p-tau231 biomarkers are increased when subtle changes of Aβ pathology are first observable.

**Novel p-tau biomarkers detect Aβ pathology more accurately in cognitively unimpaired individuals**

We conducted a receiver operating characteristic (ROC) curve analysis to test the accuracy of the novel CSF and plasma p-tau biomarkers to discriminate A+ from A− CU individuals, defined by the CSF Aβ42/40 ratio (Fig 4A) or Aβ PET (Fig 4B and C). For Aβ PET, we used as a cut-off both the visual read (usually used in the clinical setting; Fig 4B) and CL > 12 (as an early cut-off for subtle Aβ pathology; Fig 4C). Table 3 shows the areas under the curve (AUCs) for each of the p-tau biomarkers. Using any of these three definitions of A+, the novel CSF p-tau biomarkers (N-p-tau181, N-p-tau217 or Mid-p-tau231) had statistically significant higher predictive accuracies than CSF Mid-p-tau181 or t-tau (DeLong test; Table 3; Fig 4A–C). Interestingly, CSF Mid-p-tau231 had a significantly higher, albeit mild, AUC than CSF N-p-tau181 and CSF N-p-tau217 to discriminate A+ from A− CU individuals as defined by CSF Aβ42/40 or Aβ PET CL12 (Table 3). For plasma N-p-tau181, however, the AUCs to discriminate between A+ and A− in CU individuals were significantly lower (Table 3; Fig 4) than its CSF counterpart, CSF N-p-tau181. Yet, plasma N-p-tau181 had similar AUCs to those of plasma NfL (except for Aβ PET CL12 where plasma N-p-tau181 had a significantly higher AUC).

**Associations between p-tau biomarkers**

We tested the correlations between the CSF p-tau biomarkers (Fig EV2). All CSF p-tau species had a significant and strong or very

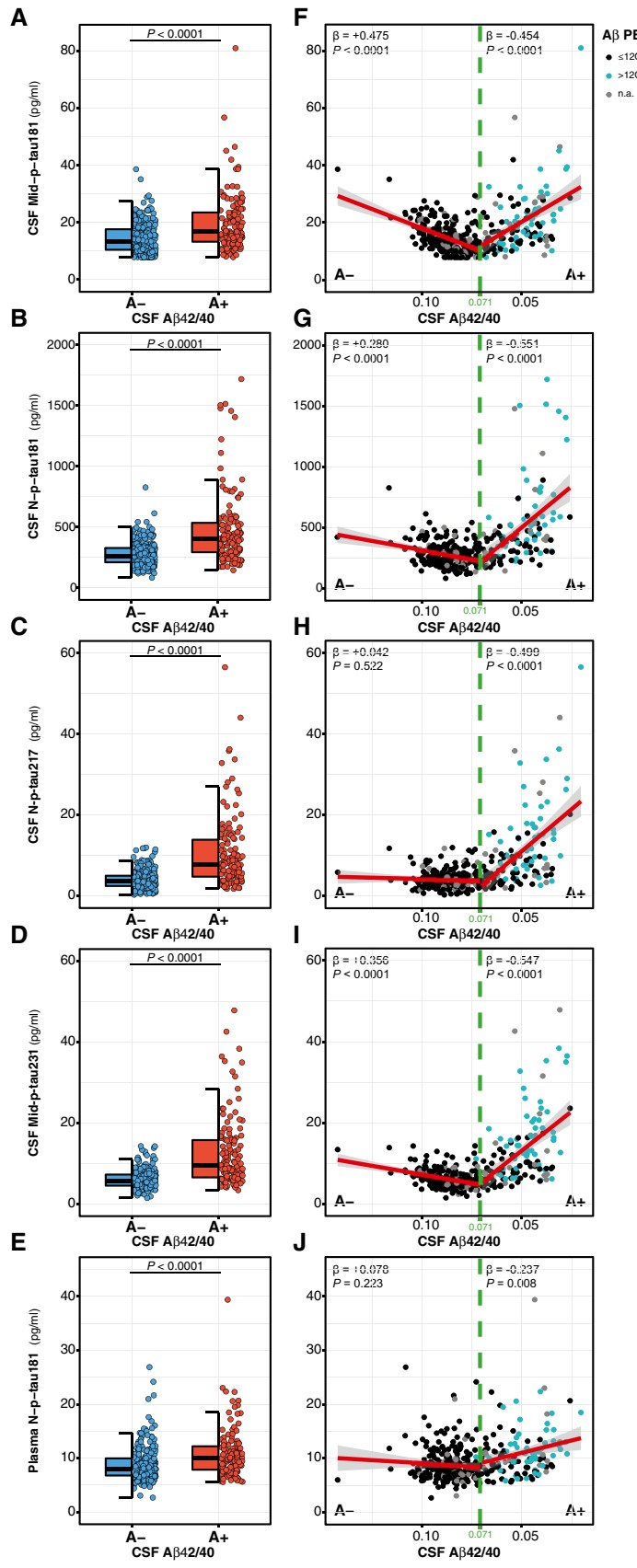

Figure 2.

**Figure 2. Effect of Aβ pathology (CSF Aβ42/40) on p-tau biomarkers.**

A–E Dot and box-plot comparing each of the p-tau biomarker between the Aβ-negative (A−; blue; n = 250) and the Aβ-positive (A+; red; n = 131) groups. Aβ positivity was defined as CSF Aβ42/40 ratio < 0.071. The box-plots depict the median (horizontal bar), interquartile range (IQR, hinges) and 1.5 × IQR (whiskers). P-values were assessed by a one-way ANCOVA adjusted for age and sex.

F–J Scatter plots depicting the changes between each p-tau biomarker as a function of CSF Aβ42/40. The horizontal axes directions were inverted; lower CSF Aβ42/40 ratio reflects higher Aβ pathology. For each Aβ status group, we computed the standardized regression coefficients (β) and the P-values, adjusted for age and sex. The solid lines indicate the regression line and the 95% confidence intervals for each of the Aβ status groups. The dashed green lines indicate the CSF Aβ42/40 cut-off. Participants were also colour-coded based on the Aβ PET CL scale (≤ 12CL, black; > 12CL, turquoise; Aβ PET non-available, grey).

Data information: Abbreviations: CSF, cerebrospinal fluid; Mid, mid-region; N, N-terminal; n.a. non-available; p-tau, phosphorylated tau.

strong correlation between them, as shown by Spearman's correlation coefficients ($r_s$) above 0.60. Most of these correlations were higher in the A+ than the A− group (Fig EV2). Unlike the relationship between CSF p-tau biomarkers, the correlation between CSF and plasma N-p-tau181 was weaker but still significant (Fig EV3). Of note, the correlation of CSF and plasma N-p-tau181 was higher in the A+ ($r_s$ = +0.353, $P$ < 0.0001) than in the A− group ($r_s$ = +0.191, $P$ = 0.003), similar to what has been reported in previous studies (Janelidze et al, 2020a).

We next asked whether the novel CSF p-tau biomarkers were already increased before CSF Mid-p-tau181, used herein as a gold standard biomarker of tau pathology, becomes positive. For these purposes, we applied the AT classification [A as defined by CSF Aβ42/40 and T by the CSF Mid-p-tau181, using previously published cut-offs (Milà-Alomà et al, 2020)]. Interestingly, we found that participants classified as A+T− [namely, Preclinical Alzheimer's pathological change (Jack et al, 2018)] had a significant increase in all the novel p-tau biomarkers (Fig 5A–D). Specifically, the increase in the A+T− group compared to the A−T− group was as follows: 37.0% for CSF N-p-tau181 (Fig 5A), 83.5% for CSF N-p-tau217 (Fig 5B), 59.3% for CSF Mid-p-tau231 (Fig 5C) and 15.8% for plasma N-p-tau181 (Fig 5D). In comparison, the CSF Mid-p-tau181

mean only had an increase of 12.5% and, by definition, all CSF Mid-p-tau181 values were below the cut-off for tau positivity. Noticeably, the levels of CSF N-p-tau217 in the A−T+ group (namely, CU non-Alzheimer's pathologic change) were not increased (Fig 5B). Altogether, these data suggest that CSF N-p-tau181, N-p-tau217 and Mid-p-tau231 are very sensitive biomarkers that capture the earliest stages of the Alzheimer's continuum.

### Association of p-tau biomarkers with neurodegeneration biomarkers

We next investigated the association of the CSF and plasma p-tau biomarkers with CSF NfL, as a biomarker of neurodegeneration (Fig EV4). In the whole sample, both CSF and plasma p-tau biomarkers were significantly associated with CSF NfL ($P$ < 0.0001 for all CSF p-tau biomarkers and $P$ = 0.025 for plasma N-p-tau181). However, these associations differed between the A+ and A− groups. In the A+ one, there were higher increases of CSF p-tau biomarkers per unit of CSF NfL than in the A− group (as shown by the different standardized regression coefficients). Plasma N-p-tau181 showed a tendency to be associated with CSF NfL in the A+ group ($P$ = 0.053) but not in the A− one (Fig EV4).

**Table 2. Association of p-tau biomarkers and Aβ biomarker (either CSF Aβ42/40 or Aβ PET Centiloid scale) stratifying by Aβ status.**

| | CSF Aβ42/40 (n = 381) | | | | | | | Aβ PET (n = 329) | | | | | | |
|---|---|---|---|---|---|---|---|---|---|---|---|---|---|---|
| | A− (n = 250) | | | A+ (n = 131) | | | "Aβ42/40 × Aβ status" interaction | A− (n = 287) | | | A+ (n = 42) | | | "Aβ PET × Aβ status" interaction |
| | β (SE) | P | eta² | β (SE) | P | eta² | P | β (SE) | P | eta² | β (SE) | P | eta² | P |
| CSF Mid-p-tau181 | +0.475 (0.056) | <0.0001* | 0.226 | −0.454 (0.075) | <0.0001* | 0.226 | <0.0001* | +0.074 (0.059) | 0.210 | 0.006 | +0.519 (0.135) | 0.0005* | 0.280 | 0.024* |
| CSF N-p-tau181 | +0.280 (0.061) | <0.0001* | 0.079 | −0.551 (0.073) | <0.0001* | 0.312 | <0.0001* | +0.231 (0.058) | <0.0001* | 0.054 | +0.668 (0.143) | <0.0001* | 0.379 | 0.049* |
| CSF N-p-tau217 | +0.042 (0.065) | 0.522 | 0.002 | −0.499 (0.072) | <0.0001* | 0.275 | <0.0001* | +0.266 (0.058) | <0.0001* | 0.072 | +0.711 (0.104) | <0.0001* | 0.552 | 0.641 |
| CSF Mid-p-tau231 | +0.356 (0.060) | <0.0001* | 0.127 | −0.547 (0.073) | <0.0001* | 0.310 | <0.0001* | +0.271 (0.057) | <0.0001* | 0.073 | +0.714 (0.116) | <0.0001* | 0.506 | 0.139 |
| Plasma N-p-tau181 | +0.078 (0.064) | 0.223 | 0.006 | −0.237 (0.088) | 0.008* | 0.056 | 0.004* | +0.217 (0.058) | 0.0002* | 0.048 | +0.432 (0.157) | 0.009* | 0.170 | 0.759 |

CSF, cerebrospinal fluid; Mid, mid-region; N, N-terminal; PET, positron emission tomography; p-tau, phosphorylated tau.
For each p-tau biomarker, we computed the linear regression standardized coefficients (β) and standard errors (SE) as a function of a Aβ pathology biomarker (either CSF Aβ42/40 or Aβ PET Centiloid scale). The analyses were performed after stratifying for Aβ-negative (A−) and Aβ-positive (A+) groups, as defined by CSF Aβ42/40 ratio or Aβ PET visual read, respectively. We report the eta-squared (eta²) as a measure of the effect size. We also computed the P-value for the interaction term "Aβ biomarker × Aβ status". All analyses were adjusted by age and sex. Among the 331 participants with Aβ PET, two did not have visual read assessment.
*Significant differences.

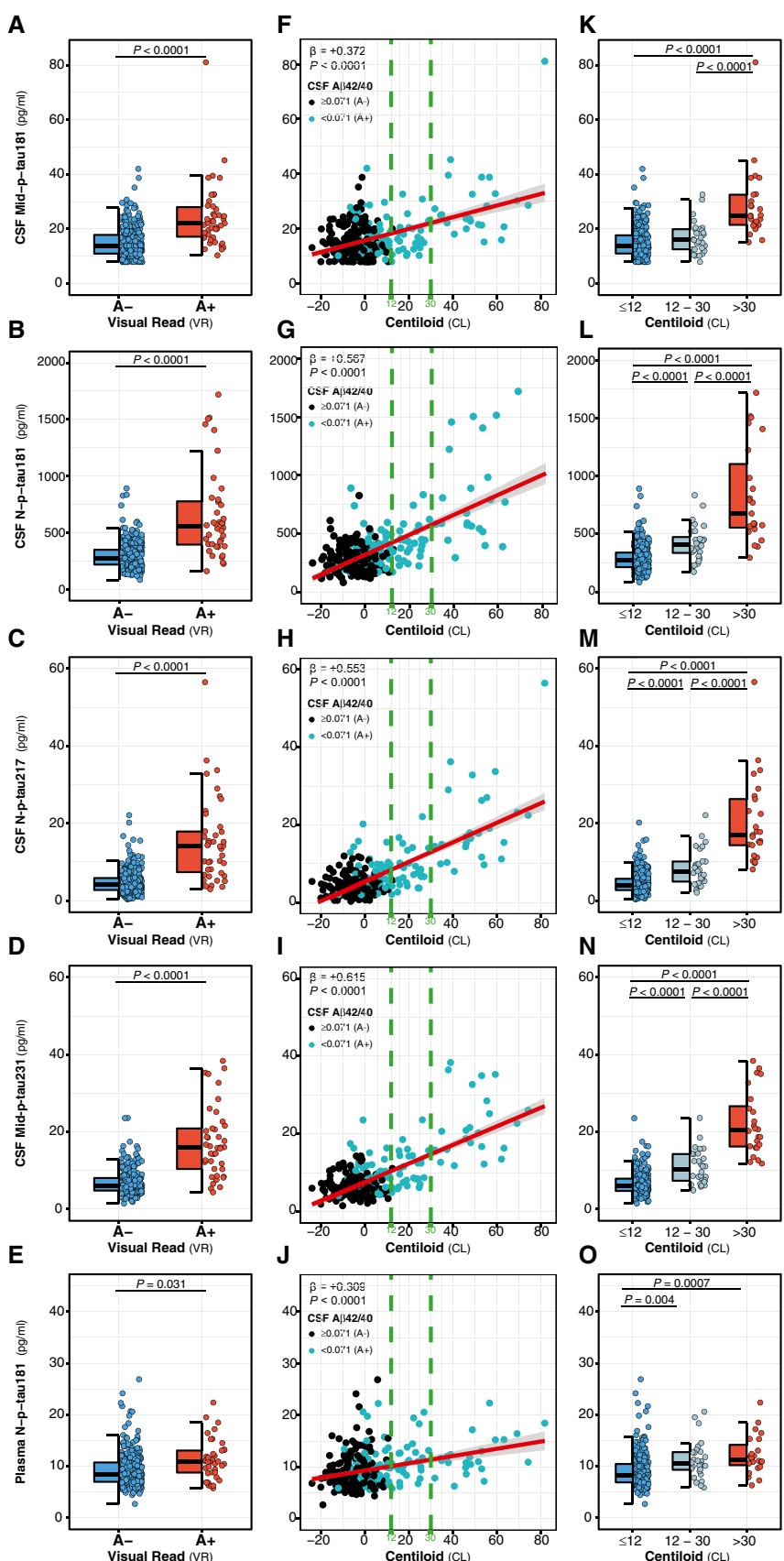

**Figure 3.**

**Figure 3.  Effect of Aβ pathology (Aβ PET) on p-tau biomarkers.**

A–E   Dot and box-plot comparing each of the p-tau biomarker between the Aβ-negative (A−; blue; n = 287) and the Aβ-positive (A+; red; n = 42) groups. Aβ positivity was defined with Aβ PET visual read. The box-plots depict the median (horizontal bar), interquartile range (IQR, hinges) and 1.5 × IQR (whiskers). P-values were assessed by a one-way ANCOVA adjusted by age and sex.

F–J   Scatter plots depicting the changes between each p-tau biomarker as a function of Aβ PET Centiloids (CL). The standardized regression coefficients (β) and the P-values were computed using a linear model adjusting for age and sex. The solid lines indicate the regression line and the 95% confidence intervals. The dashed green lines indicate the CL12 and CL30 cut-offs. Participants were also colour-coded based on the CSF Aβ42/40 ratio (A−, black; A+, turquoise).

K–O   Dot and box-plots depicting comparison between each of the p-tau biomarker between Centiloid scale groups: (i) ≤ 12CL (blue; n = 278), (ii) 12–30CL (subthreshold Aβ pathology group; grey; n = 28), (iii) >30CL (red; n = 25). The box-plots depict the median (horizontal bar), interquartile range (IQR, hinges) and 1.5 × IQR (whiskers). P-values were assessed by a one-way ANCOVA adjusted by age and sex, followed by a Bonferroni-corrected *post hoc* pairwise comparison.

Data information: Abbreviations: CSF, cerebrospinal fluid; Mid, mid-region; N, N-terminal; p-tau, phosphorylated tau.

## Modelling of p-tau biomarker changes across the preclinical Alzheimer's *continuum*

Finally, we applied a robust local weighted regression method to model the trajectories of the p-tau biomarkers across the preclinical Alzheimer's *continuum* using CSF Aβ42/40 as a proxy of disease progression. The resulting plots are shown in Fig 6, which depict the z-scores changes of each of the p-tau biomarkers after correcting for age and sex and standardizing by the standard deviations (SD) of the A−T− group. We defined biomarker abnormality as z-scores > 2, corresponding to 2 SD of the adjusted values of the A−T− group. For comparison purposes, we also added the well-established AD-related biomarkers CSF t-tau and plasma NfL.

Using CSF Aβ42/40 as a proxy of disease duration, we observed that the novel p-tau biomarkers (either N-p-tau181, N-p-tau217 or Mid-p-tau231) had more pronounced increases in A+ participants (as shown in the slopes in Fig 6 and consistent with the standardized regression coefficients (β) in A+ participants computed in Table 2). All the novel p-tau biomarkers surpassed the 2 *z*-scores levels (depicted with a horizontal dashed line in Fig 6) compared with their baseline levels (used herein as a definition of abnormal levels) within the preclinical stage of the Alzheimer's *continuum*. Specifically, Mid-p-tau231 surpassed the 2 *z*-scores at a corresponding CSF Aβ42/40 ratio of 0.044, N-p-tau181 at CSF Aβ42/40 of 0.037 and N-p-tau217 at CSF Aβ42/40 of 0.035. In comparison, Mid-p-tau181 reached the 2 *z*-scores at a CSF Aβ42/40 of 0.026. Note that the higher the CSF Aβ42/40 ratio, the earlier it is in the Alzheimer's *continuum*. In contrast, CSF t-tau and CSF NfL did not reach the 2 *z*-scores and their increases were less pronounced (Fig 6). The increase of plasma N-p-tau181 across the preclinical stage of the Alzheimer's *continuum* was milder than that of the CSF N-p-tau181 (N-p-tau181 in the A+ group, CSF: β = −0.551, P < 0.0001; plasma: β = −0.237, P = 0.008 in plasma; Table 2). However, plasma N-term-p-tau181 reached an increase of 1 z-scored in the preclinical stage of the Alzheimer's *continuum* (Fig 6), higher than the increase observed in plasma NfL (that reached 0.5 z-scores). We repeated this analysis using Aβ PET as a proxy of disease progression (Fig EV5), and we observed a similar pattern, with the novel CSF p-tau biomarkers reaching the 2 z-scores following this order: CSF Mid-p-tau231, N-p-tau181 and N-p-tau217, followed by Mid-p-tau181 and t-tau. In other words, the novel p-tau biomarkers become abnormal with a lower amount of Aβ pathology as measured by Aβ PET uptake.

In order to further corroborate our hypothesis that the novel p-tau biomarkers increase very early in the *continuum*, we determined at what point of the disease progression these biomarkers modify their association with CSF Aβ42/40. Specifically, we computed the levels of CSF Aβ42/40 at which the levels of p-tau biomarkers present an inflection point. A higher level of CSF Aβ42/40 ratio (as a proxy of disease duration) corresponding to a particular p-tau biomarker inflection point would indicate an earlier increase of that p-tau biomarker in the *continuum* of the disease. We also computed the *z*-score change of the CSF Aβ42/40 ratio matching each inflection point (in brackets). Considering this, we found that the inflection point of CSF N-p-tau217 was at CSF Aβ42/40 = 0.084 (Aβ42/40 *z*-score = 0.28), and CSF Mid-p-tau231 and CSF N-p-tau181 at CSF Aβ42/40 = 0.082 (Aβ42/40 *z*-score = 0.51), whereas the inflection point of CSF Mid-p-tau181 and t-tau was at CSF Aβ42/40 = 0.080 (Aβ42/40 *z*-score = 0.75). In other words, the novel CSF p-tau biomarkers modify their association with CSF Aβ42/40 when this ratio is only 0.082–0.084 (not yet reaching the cut-off of positivity set at 0.071), and the CSF Aβ42/40 ratio was only between 0.28–0.51 *z*-scores higher than its adjusted values of the A− group. Regarding plasma biomarkers, plasma N-p-tau181 had its inflection point at CSF Aβ42/40 = 0.080, earlier in the *continuum* than plasma NfL, which was at CSF Aβ42/40 = 0.076.

Altogether, we demonstrate using different approaches that these novel p-tau biomarkers increase with subtle and incipient Aβ changes, most likely reflecting very early changes in the *continuum* of the disease.

## Discussion

We investigated novel CSF and plasma p-tau biomarkers in the preclinical stage of the Alzheimer's *continuum*. We hypothesized that they change when subtle changes in Aβ are detectable. Our main results show that, compared to the widely used CSF Mid-p-tau181 or t-tau biomarkers, the novel CSF N-p-tau181, N-p-tau217 and Mid-p-tau231: (i) increase more prominently early in the *continuum*, when using CSF Aβ42/40 ratio as a proxy of disease duration, (ii) increase with subthreshold levels of Aβ pathology, (iii) can more accurately differentiate Aβ-positive (A+) CU individuals from those that are Aβ-negative (A−). Moreover, we also provide evidence that the novel plasma N-p-tau181 significantly increases in the preclinical stage of the Alzheimer's *continuum*, albeit the magnitude of that increase is not as high as that of the CSF p-tau biomarkers.

Tau can be phosphorylated at multiple sites, and several of these sites have been investigated as biomarkers for AD (Ishiguro *et al*, 1999; Kohnken *et al*, 2000; Hu *et al*, 2002; Buerger *et al*, 2002b;

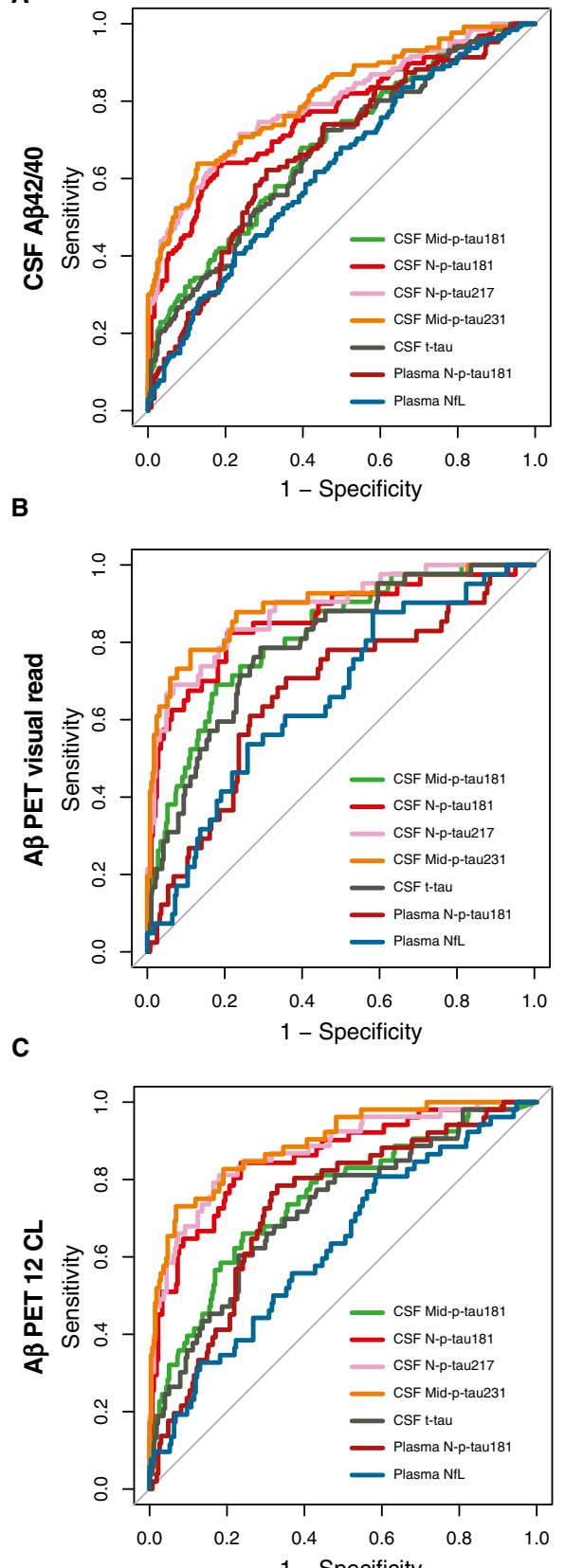

**Figure 4. Discrimination of cognitively unimpaired Aβ-positive from Aβ-negative individuals by p-tau biomarkers.**

A–C ROC analysis was performed to test the accuracy to discriminate between Aβ-positive (A+) from Aβ-negative (A−) individuals. Aβ positivity was defined as CSF Aβ42/40 < 0.071 (A), Aβ PET-positive visual read (B) or Aβ PET Centiloid (CL) > 12 (C). Abbreviations: CSF, cerebrospinal fluid; Mid, mid-region; NfL, neurofilament light; N, N-terminal; p-tau, phosphorylated tau; t-tau, total tau.

Hampel *et al*, 2004; Singer *et al*, 2009; Meredith *et al*, 2013; Russell *et al*, 2016; Janelidze *et al*, 2020b). Tau phosphorylated at threonine-181 (p-tau181) is the most commonly used as CSF biomarker but new data indicate that other phosphorylation sites may be at least as useful as p-tau181. Tau phosphorylated at threonine-217 (p-tau217) can be readily detected in CSF, and it is increased in AD (preprint: Barthélemy *et al*, 2017; Barthélemy *et al*, 2019; Sato *et al*, 2018). Recently, two independent studies showed that CSF p-tau217 may be better than CSF p-tau181 to detect AD (Barthélemy *et al*, 2020a; Janelidze *et al*, 2020b). In a recent study in the BioFINDER cohort, CSF p-tau217 measured by an immunoassay distinguished AD dementia from other neurodegenerative diseases better than CSF p-tau181 and, what is more, correlated better with tau PET and showed higher longitudinal changes than CSF p-tau181. This study also included 40 CU Aβ-positive participants who had significantly higher CSF p-tau217 than a group of 25 CU Aβ-negative participants (Janelidze *et al*, 2020b). A second study using immunoprecipitation mass spectrometry demonstrated that CSF p-tau217 levels were six times higher in AD compared to neurological controls, while CSF p-tau181 only showed a 1.3-fold increase. Importantly, that difference was replicated in a validation cohort that also comprised prodromal AD and CU individuals who were Aβ-positive (Barthélemy *et al*, 2020a). In our study, we extend these findings demonstrating in a larger cohort, including a high number of individuals in the preclinical stage of the Alzheimer's *continuum*, that CSF N-p-tau217 increases very early in the disease, even with very subtle Aβ changes. This increase was observed even in the group of participants who were classified as A+ but T− (i.e. Preclinical Alzheimer's pathological change), as defined by the CSF Mid-p-tau181 assay, suggesting that CSF N-p-tau217 may be more sensitive to detect early changes. Furthermore, CSF N-p-tau217 was not increased in the A−T+ groups (that is, the CU non-Alzheimer's pathologic change, as defined also by CSF Mid-p-tau181), which also suggests a higher specificity of CSF N-p-tau217 for AD. This probable higher specificity is in line with recent studies demonstrating that plasma p-tau217 detects AD pathology with a very high accuracy (Palmqvist *et al*, 2020; Barthélemy *et al*, 2020b). An unexpected and interesting finding was that CSF N-p-tau217 was slightly higher in women than men, which is consistent with studies that show that women have higher levels of tau pathology in pathological and PET studies (Filon *et al*, 2016; Buckley *et al*, 2019). Gender and sex differences in tau and other biomarkers do need further investigation to develop a precision medicine approach in AD (Ferretti *et al*, 2018).

CSF p-tau231 has been measured during the last two decades using different techniques, and it has been proven to be useful to detect AD (prodromal or dementia) (Kohnken *et al*, 2000; Arai *et al*, 2000; Blennow *et al*, 2001; De Leon *et al*, 2002; Buerger *et al*, 2002a;

**Table 3. ROC analyses to discriminate cognitively unimpaired Aβ-positive from Aβ-negative individuals.**

| | AUC (95% CI) CU Aβ+ vs. Aβ− | | |
| --- | --- | --- | --- |
| | CSF Aβ42/40 (A−: 250; A+: 131) | Aβ PET visual read (A−: 287; A+: 42) | Aβ PET CL12 (A−: 278; A+: 53) |
| CSF Mid-p-tau181 | 0.683 (0.627–0.739) | 0.814 (0.748–0.881) | 0.748 (0.673–0.824) |
| CSF N-p-tau181 | 0.763 (0.710–0.816)[a] | 0.858 (0.787–0.928)[c] | 0.855 (0.794–0.916)[a] |
| CSF N-p-tau217 | 0.794 (0.744–0.844)[a] | 0.883 (0.825–0.941)[g] | 0.875 (0.819–0.931)[a] |
| CSF Mid-p-tau231 | 0.809 (0.761–0.856)[a,b] | 0.894 (0.833–0.954)[a] | 0.894 (0.845–0.943)[a,j] |
| CSF t-tau | 0.669 (0.612–0.726)[c,d,e,f] | 0.793 (0.725–0.862)[b,e,f,g] | 0.722 (0.646–0.799)[a,d,e,f] |
| Plasma N-p-tau181 | 0.670 (0.612–0.728)[b,e,f] | 0.671 (0.581–0.762)[d,e,f,g,h] | 0.729 (0.656–0.801)[b,e,f] |
| Plasma NfL | 0.626 (0.567–0.685)[d,e,f] | 0.660 (0.574–0.745)[d,e,f,g,i] | 0.628 (0.545–0.710)[d,e,f,g,i,k] |

Aβ42, amyloid-β 42; Aβ40, amyloid-β 40; AUC, area under the curve; CI, confidence interval; CL, Centiloid; CSF, cerebrospinal fluid; Mid, mid-region; N, N-terminal; NfL, neurofilament; p-tau, phosphorylated tau; t-tau, total tau; PET, positron emission tomography.
ROC analyses to test whether each p-tau biomarker discriminates between Aβ-positive (A+) and Aβ-negative individuals (A−), as defined by the CSF Aβ42/40 ratio, Aβ PET visual read or Aβ PET using a cut-off of CL12. We also included CSF t-tau and plasma NfL for comparison. Aβ PET was performed in 331 participants, and Centiloid scale was available in all of them; two participants did not have visual read assessment. The exact P-values are shown in Appendix Table S1. AUCs differences were tested using the DeLong test. The significant differences were as follows:
[a]$P < 0.001$ compared to CSF Mid-p-tau181.
[b]$P < 0.01$ compared to CSF N-p-tau181.
[c]$P < 0.05$ compared to CSF Mid-p-tau181.
[d]$P < 0.001$ compared to CSF N-p-tau181.
[e]$P < 0.001$ compared to CSF N-p-tau217.
[f]$P < 0.001$ compared to CSF Mid-p-tau231.
[g]$P < 0.01$ compared to CSF Mid-p-tau181.
[h]$P < 0.01$ compared to CSF t-tau.
[i]$P < 0.05$ compared to CSF t-tau.
[j]$P < 0.05$ compared to CSF N-p-tau181.
[k]$P < 0.05$ compared to plasma N-p-tau181.

Hampel et al, 2004, 2005; Buerger et al, 2009; Glodzik-Sobanska et al, 2009; Meredith et al, 2013; Spiegel et al, 2015; Wang et al, 2016; Öhrfelt et al, 2016; Kiđemet-Piskač et al, 2018; Babić Leko et al, 2018; Santos et al, 2019). To the best of our knowledge, our study is the first one to investigate CSF p-tau231 in the preclinical Alzheimer's continuum and the results are striking. Among the CSF p-tau biomarkers studied, CSF Mid-p-tau231 is the one that increased most prominently. This result is consistent with neuropathological findings demonstrating that p-tau231 is present in pre-neurofibrillary tangles, prior to overt filament formation (Augustinack et al, 2002; Luna-Muñoz et al, 2005, 2007). Furthermore, in a pathological study discriminating insoluble NFT from soluble oligomeric and detergent-soluble tau, phosphorylation at threonine-231 was one of the consistently increased tau post-translational modifications that discriminated Braak stage 0–I from III–IV (Ercan-Herbst et al, 2019). Thus, based on our data, CSF Mid-p-tau231 is one of the most promising biomarkers for preclinical Alzheimer.

It is important to highlight that most commonly used t-tau and p-tau assays (including INNO-BIA AlzBio3, INNOTEST, Lumipulse and Elecsys) use antibodies against the mid-region of the protein. However, other tau fragments are present in the CSF that are not detected by these assays (Meredith et al, 2013). Here, we used two assays targeting p181 but differed in the tau fragment that was measured. While the Elecsys assay measures mid-region tau forms (as most of the currently used tau assays do), the Simoa one measures N-terminal forms (Fig EV1). In all the analyses, the N-p-tau181 assay had a higher increase with incipient Aβ pathology than the Mid-p-tau181 assay. Alternatively, these differences might

be related to the use of different antibodies for p-tau181 in the different assays (see Fig EV1). Nevertheless, these results suggest that both the phosphorylation site and the fragment of tau measured are important to consider when developing assays and highlight that it may be important in future studies to specify whether a tau assay detects a mid-region or an N-terminal fragment.

CSF p-tau (either p-tau181 or other phosphorylation sites) have usually been considered as a marker of NFT pathology. However, this notion has been challenged in the last years by different means. First of all, the agreement between CSF p-tau and tau PET [that binds to NFT (Chien et al, 2013; Lowe et al, 2016; Marquié et al, 2017)] is inconsistent across studies conducted in CU individuals (Brier et al, 2016; Gordon et al, 2016; Chhatwal et al, 2016; Mattsson et al, 2017; Schöll et al, 2019). In line with this, studies on the timing of the biomarker's changes in sporadic or autosomal-dominant AD suggest that CSF p-tau increases earlier in the disease than tau PET (Bateman et al, 2012; Toledo et al, 2013; Johnson et al, 2016; Quiroz et al, 2018; Gordon et al, 2019; Jack et al, 2019; Mattsson et al, 2019). Second, there is now clear evidence of an active secretion of tau in both physiological and pathological conditions, indicating that p-tau does not merely result from the passive leakage of NFT of dead neurons to the CSF (Kim et al, 2010; Karch et al, 2012; Simón et al, 2012; Plouffe et al, 2012; Saman et al, 2012; Chai et al, 2012; Pooler et al, 2013; Maia et al, 2013; Kanmert et al, 2015; Sato et al, 2018). Likewise, neuronal death (as measured by fluid or neuroimaging biomarkers) also occurs later than CSF p-tau changes in the disease continuum. Furthermore, the phosphorylation pattern of tau differs in the brain and CSF of individuals with and without AD (Barthélemy et al, 2019) and

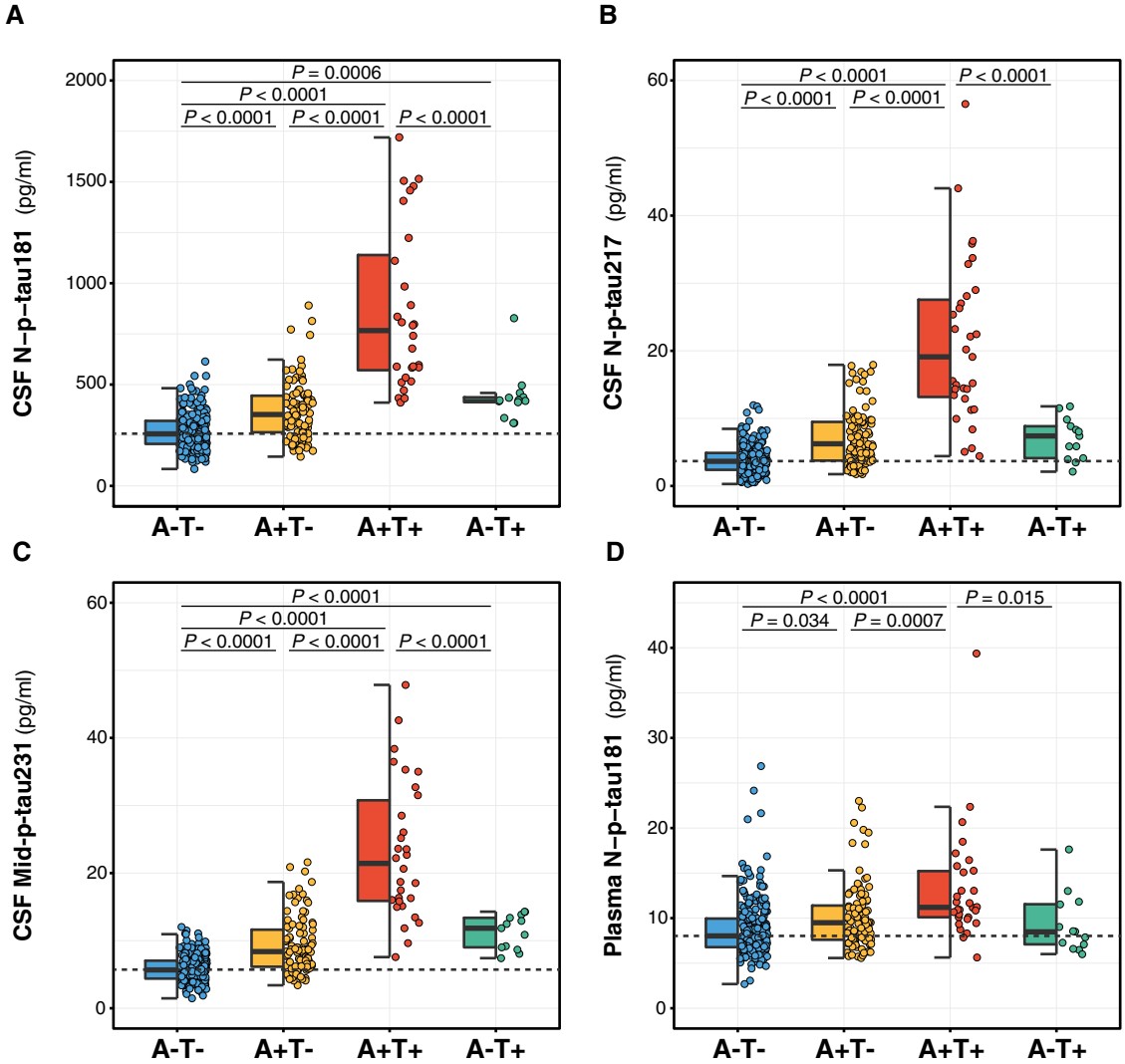

**Figure 5. Comparison of the p-tau biomarkers between AT groups.**

A–D Dot and box-plot showing the levels of each p-tau biomarker in each of the AT groups. Aβ-positive (A+) was defined by a CSF Aβ42/40 < 0.071 and Tau-positive (T+) by an Elecsys CSF Mid-p-tau181 > 24 pg/ml. The box-plots depict the median (horizontal bar), interquartile range (IQR, hinges) and 1.5 × IQR (whiskers). The horizontal dashed line indicates the median of the p-tau biomarker in the A−T− group. *P*-values were assessed by a one-way ANCOVA adjusted by age and sex followed by Bonferroni-corrected *post hoc* pairwise comparisons. Abbreviations: CSF, cerebrospinal fluid; Mid, mid-region; N, N-terminal; p-tau, phosphorylated tau.

dynamically changes during the course of the disease (Barthélemy *et al*, 2020c). The results of our study also support the idea that there might be tau pathophysiological processes (e.g. increased proteolytic cleavage, phosphorylation, active secretion) that are relevant for the disease and occur before the actual aggregation of p-tau into NFT. We add to the existing literature the fact that novel CSF p-tau biomarkers (N-p-tau181, N-p-tau217 or Mid-p-tau231) are closely linked to Aβ pathology in sporadic AD, even when this Aβ pathology is very subtle. To note, the cut-off used here for CSF Aβ42/40 has been optimized to detect preclinical Alzheimer and is higher (and thus more sensitive) than cut-offs with high specificity usually used for diagnostic purposes in symptomatic AD (Milà-Alomà *et al*, 2020). Likewise, we used the cut-off of CL > 12 (Salvadó *et al*, 2019), which reflects very mild Aβ changes, and it is

considerably lower than the CL values found when Aβ pathology is well-established in symptomatic AD (Leuzy *et al*, 2016; Rowe *et al*, 2017; La Joie *et al*, 2019; Salvadó *et al*, 2019; Amadoru *et al*, 2020). Even with these very early cut-offs, we were able to detect significant changes in these novel CSF p-tau biomarkers. Although these data are highly suggestive of a link between early Aβ pathology and active secretion of specific p-tau species, we are cognizant that this is an observational study and we cannot hence draw a cause-effect conclusion. Nevertheless, studies in transgenic mice models of AD have suggested that specific changes in p-tau metabolism followed those of Aβ (Maia *et al*, 2013; Mattsson-Carlgren *et al*, 2020).

One of the most exciting breakthroughs in the AD field has been the recent development of blood tau phosphorylated at site 181 (p-tau181) as a biomarker for AD (Tatebe *et al*, 2017; Mielke *et al*, 2018;

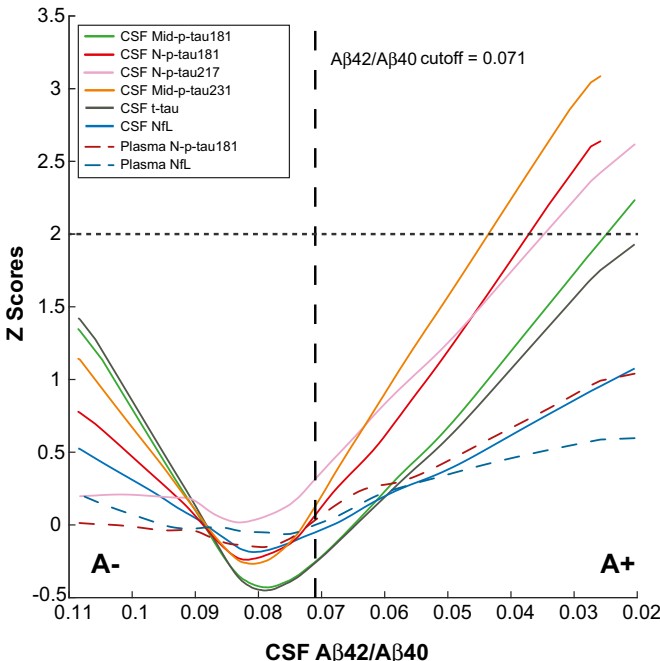

**Figure 6. Trajectories of the p-tau biomarkers in preclinical Alzheimer.**

The graphs represent the z-scores changes of each CSF biomarker as a function of CSF Aβ42/40 ratio (as proxy of disease progression) using a robust local weighted regression method. The z-scores were calculated using the mean and the SD of each CSF biomarker in the A−T− group as a reference. The solid lines depict the trajectory of each CSF biomarker. The dashed lines depict the trajectories of the plasma biomarkers. The vertical black dashed line indicates the CSF Aβ42/40 cut-off for A+. Note that the CSF p-tau biomarkers reach the 2 z-scores (depicted with an horizontal dashed line) with the following sequence: Mid-p-tau231, N-p-tau181, N-p-tau217, Mid-p-tau181 and t-tau. Abbreviations: CSF, cerebrospinal fluid; Mid, mid-region; NfL, Neurofilament light; N, N-terminal; p-tau, phosphorylated tau; t-tau, total tau.

Karikari *et al*, 2020; Thijssen *et al*, 2020; Janelidze *et al*, 2020a). Plasma p-tau181 can discriminate with an AUC that ranges from 82 to 98% symptomatic AD from other neurological diseases. Moreover, plasma p-tau181 can accurately discriminate between Aβ PET-positive and Aβ PET-negative individuals in different stages of the Alzheimer's *continuum* (AUC ranging from 76 to 88%). Importantly, plasma p-tau181 has also a high agreement with AD pathological diagnosis, predicts progression to cognitive impairment and can be applied in the primary care to screen for AD cases. These studies have also showed that levels of plasma p-tau181 start to increase in preclinical AD, although the magnitude of those increases is not as high as those observed in symptomatic cases. Our group showed, using the same assay, that plasma N-p-tau181 has an AUC of 81% to discriminate CU individuals who were Aβ-positive from those who were Aβ-negative (Karikari *et al*, 2020). Previously, Mielke *et al* (2018) showed an AUC of 70.4% to predict elevated Aβ PET in CU individuals. In the present study, we confirm this increase in blood p-tau181 in a preclinical cohort including 381 participants and, remarkably, either using CSF Aβ42/40 or Aβ PET to detect preclinical Alzheimer. The present study has two unique features that make a significant contribution. First, our participants are middle-age, when Alzheimer's pathology most

likely starts but biomarker levels may be too subtle to detect. Second, most of our preclinical Alzheimer's participants are in the earliest stage of the *continuum*; 76% of those within the *continuum* were A+ but still T− (as defined by the CSF Mid-p-tau181 assay) and also including a considerable number of individuals with subthreshold levels Aβ pathology as defined by the Aβ PET CL scale. Considering this, one would expect much lower increases in blood N-p-tau181 in ALFA+ compared to other studies, since participants here are younger and in the very early phases of incipient Aβ pathology. Yet, we show here that these individuals already have mild but significant increases in plasma N-p-tau181. The AUC to differentiate Aβ-positive CU from Aβ-negative ranges from 0.670 to 0.729 (depending on CSF Aβ42/40 or Aβ PET definition). These AUCs preclude, at the moment, its use as a diagnostic tool for preclinical Alzheimer, but it may open the door to its use as a screening tool to enrich interventional or observational studies focused on preclinical Alzheimer, maybe in combination with refined plasma Aβ tests.

This study is not free of limitations. First, it is a cross-sectional analysis and longitudinal studies are needed to confirm changes in p-tau biomarkers during the course of the disease and assess their capacity to predict cognitive impairment. Second, we used different analytical platforms and antibodies to measure these biomarkers, which could influence the performance to identify early tau pathology in the disease *continuum*. In this regard, it is worth-mentioning that the biomarker with the most prominent change early in the disease was Mid-p-tau231, which was measured with a conventional colorimetric ELISA, instead of the ultrasensitive Simoa or Elecsys platforms, thus suggesting that the analyte was more important than the assay platform. Third, there is no tau PET available, the neuroimaging biomarker of tau pathology. Nonetheless, tau PET reflects neurofibrillary tangle deposition, while our data favour the idea that these novel p-tau assays reflect tau metabolism changes prior to tangle formation. Finally, our study did not include plasma p-tau217. While our manuscript was being reviewed, two studies were published showing the high value of this biomarker to identify AD (Palmqvist *et al*, 2020; Barthélemy *et al*, 2020b).

The major strengths of our study include the following: First, study participants were part of a very well-characterized cohort of CU individuals recruited at one centre according to a standardized protocol; the high prevalence of preclinical Alzheimer's pathological change (A+T−, as defined by the core CSF biomarkers) suggests the cohort is ideal to study early molecular changes in AD. Second, sensitive, robust and reproducible biomarker assays were used. Third, the combination in the same study of five different tau assays and, for N-p-tau181, both CSF and plasma measurements is, to the best of our knowledge, unique.

In conclusion, we show here that novel p-tau biomarkers, namely CSF and plasma N-p-tau181, CSF N-p-tau217 and CSF Mid-p-tau231 increase early in the preclinical stage of the Alzheimer's *continuum*, probably in response to emerging Aβ pathology. Our results have important practical implications for the design of clinical trials in preclinical Alzheimer. They support the idea of therapeutically targeting tau (or specific tau species) very early in the disease, as soon as Aβ pathology arises and presumably before neurofibrillary tangle deposition. Moreover, these novel p-tau biomarkers can potentially be used as target engagement biomarkers in tau-targeted therapies and also to test in proof-of-concept clinical trials the downstream effects of anti-Aβ treatments.

# Materials and Methods

### ALFA participants, study design and biomarker classification

This study was performed with the first consecutive 381 participants of the ALFA+ cohort, which is a nested longitudinal study of the ALFA (for ALzheimer's and FAmilies) study (Molinuevo *et al*, 2016). The ALFA cohort was established as a research platform to characterize preclinical AD in 2,743 cognitively unimpaired individuals, aged between 45 and 75 years old, and enriched for family history of AD (excluding autosomal-dominant AD). In the ALFA+ study, participants are longitudinally followed up and undergo a more comprehensive evaluation, including, but not limited to lumbar puncture, MRI and [$^{18}$F]flutemetamol Aβ PET.

Participants were classified as Aβ pathology-positive (A+) if CSF Aβ42/40 ratio < 0.071, and tau pathology-positive (T+) if CSF Mid-p-tau181 > 24 pg/ml. These cut-offs were previously derived using a two-Gaussian mixture modelling (Milà-Alomà *et al*, 2020). Each cut-off was defined as the mean plus 2 standard deviations (SD) of the non-pathologic Gaussian distribution.

### CSF and plasma collection, processing and storage

CSF collection, processing and storage in the ALFA+ study have been described previously (Milà-Alomà *et al*, 2020). In brief, lumbar puncture was performed at the intervertebral space L3/L4, L4/L5 or L5/S1 using a standard needle, between 8 am and 12 pm and participants had fasted for at least 8 h. CSF was collected into a 15-ml sterile polypropylene sterile tube (Sarstedt, Nümbrecht, Germany; cat. no. 62.554.502), aliquoted in volumes of 0.5 ml into sterile polypropylene tubes (0.5 ml Screw Cap Micro Tube Conical Bottom; Sarstedt, Nümbrecht, Germany; cat. no. 72.730.005) and immediately frozen at −80°C.

Blood samples were obtained the same day of the lumbar puncture and, therefore, in fasting conditions. Whole blood was drawn with a 20 g or 21 g needle gauge into a 10-ml ethylenediaminetetraacetic acid (EDTA) tubes (BD Hemogard 10ml; K2EDTA; cat. no. 367525). Tubes were gently inverted 5–10 times and centrifuged at 2,000 *g* for 10 min at 4°C. The supernatant was aliquoted in volumes of 0.5 ml into sterile polypropylene tubes (Sarstedt Screw Cap Micro Tube; 0.5 ml; PP; ref. 72.730.105) and immediately frozen at −80°C.

For both CSF and plasma, the time between collection and freezing was less than 30 min and the samples were kept at room temperature during their processing. The time of the storage of the samples used in the present study ranged from 6 to 38 months. Samples were shipped on dry ice from Barcelona to the Clinical Neurochemistry Laboratory, Sahlgrenska University Hospital, Mölndal (Sweden), where all the measurements were performed.

### Measurements of core AD biomarkers

CSF Mid-p-tau181 and t-tau were measured using the electrochemiluminescence immunoassays Elecsys phospho-tau(181P) CSF and total-tau CSF, respectively, on a fully automated cobas e 601 instrument (Roche Diagnostics International Ltd.). CSF Aβ42, Aβ40 and NfL were measured with the NeuroToolKit (Roche Diagnostics International Ltd.) on a cobas e 411 or e 601 instrument. Measurements of CSF Mid-p-tau181, t-tau, Aβ42, Aβ40 and NfL were previously reported (Milà-Alomà *et al*, 2020). Plasma NfL was measured using the commercial Quanterix® assay (Simoa® NF-light Kit cat. no. 103186) on a HD-X analyser following the manufacturer's instructions (Quanterix, Billerica, MA, USA). Samples were run in singlicates and using the same batch of reagents. Each assay plate included internal quality control samples with high and low plasma NfL concentrations analysed in duplicate both in the beginning and end of the plate. Quality control samples had a mean intra-assay and inter-assay coefficient of variation of less than 5 and 7%, respectively.

### Measurements of novel p-tau biomarkers

CSF and EDTA plasma samples were allowed to thaw at room temperature for an hour, vortexed at 200 rpm for 30 s. Plasma samples were additionally centrifuged at 4,000 *g* for 10 min. Samples were assayed fourteen-fold diluted in the CSF N-p-tau181 assay, four-fold diluted in the N-p-tau217 assay and two-fold diluted in the plasma N-t-pau181 assay. CSF samples were run neat in the Mid-p-tau231 assay. Internal quality control samples analysed in duplicates at the start and the end of each plate were used to determine the within- and between-run variations (Table EV3), following the recommendations of an international group of neurochemists (Andreasson *et al*, 2015).

CSF and plasma N-p-tau181 were measured on the Simoa HD-X instrument (Quanterix, Billerica, MA, USA) following previously published protocols (Karikari *et al*, 2020). Briefly, the p-tau181-specific monoclonal antibody AT270 was used as capture antibody (Goedert *et al*, 1994; Vanmechelen *et al*, 2000). The N-terminal mouse monoclonal antibody Tau12 that recognizes the N-terminal epitope 6-QEFEVMEDHAGT-18 on full-length human tau was used as the detection antibody (Horowitz *et al*, 2004). We used 20,000 beads/μl of capture antibody and 2 μg of detection biotin-conjugated antibody. Identical conditions were used for the CSF N-p-tau217 assay except that the rabbit polyclonal antibody specific for tau phosphorylated at threonine-217 (#44-744, Invitrogen, US) was used as the capture antibody (Ercan *et al*, 2017), and the assay used a 2-step Homebrew protocol (where the analyte, the bead-coated capture antibody and biotinylated detection antibody are incubated together in the first step) instead of a 3-step protocol (where the first step involves the incubation of the target analyte with the capture antibody only; detection antibody is added after washing to remove unbound analytes). Both assays used an identical calibrator, that is, full-length recombinant tau 1–441 phosphorylated *in vitro* by glycogen synthase kinase 3β (#TO8-50FN, SignalChem, Canada). The specificity of the assays for tau forms that contain the indicated analytes was verified by mass spectrometry as described before (Karikari *et al*, 2020).

CSF Mid-p-tau231 was quantified using a research ELISA assay using cis-conformational selective monoclonal antibody (ADx253, ADx NeuroSciences). New monoclonal mouse antibodies were generated using a 17-mer synthetic peptide, phosphorylated on hTau corresponding Thr231, spanning the tau region $K_{224}KVAVVR$ (pT)$PPKSPSSAK_{240}C$ as a KLH-coupled antigen. Candidate hybridomas were selected on brain extracts of AD and control brain tissue. The final cloned and purified monoclonal antibody, ADx253, was characterized on synthetic peptides spanning the region $T_{217}$ till $S_{241}$

for its affinity, its phospho-specificity and its preferred selectivity in which P232 was replaced by a Pip, to simulate cis-selectivity of ADx253 (De Vos *et al*, 2016). A phospho231-specific, cis-conformational selective colorimetric ELISA was based on ADx253 monoclonal antibody coated on the plate and detected by a biotinylated pan tau monoclonal antibody ADx205 (epitope region aa 185–195; ADx NeuroSciences). We used 80 μl of sample per well and in duplicate. The assay is calibrated using an in house designed synthetic peptide combining both antibody epitopes and having the corresponding threonine-231 phosphorylated and proline-232 replaced by a homoproline, Pip (De Vos *et al*, 2016).

Among the 381 samples, the following measurements were not available: 4 CSF N-p-tau181, 6 CSF N-p-tau217, 1 CSF Mid-p-tau231 and 11 plasma Mid-p-tau181. All measurements were performed at the Clinical Neurochemistry Laboratory, University of Gothenburg, Mölndal, Sweden, by laboratory technicians and scientists blinded to participants' clinical information.

## Amyloid-β positron emission tomography acquisition and processing

[18F]flutemetamol PET scans (referred throughout the text as Aβ PET) were available for 86.9% of the participants. Imaging acquisition and preprocessing protocols have been described previously (Salvadó *et al*, 2019). In brief, PET imaging was conducted in a Siemens Biograph mCT, following a cranial CT scan for attenuation correction. Participants were injected with 185 MBq (range 166.5–203.5 Mbq) of [18F]flutemetamol, and four frames of 5 min each were acquired 90 min post-injection.

PET processing was performed following the standard Centiloid (CL) pipeline (Klunk *et al*, 2015) using SPM12. In brief, PET images are first co-registered to their respective T1-weighted images and afterwards moved to MNI space using the normalization transformation derived from the segmentation of the T1-weighted image. PET images were intensity normalized using whole cerebellum as reference region. CL values were calculated from the mean values of the standard CL target region (http://www.gaain.org/centiloid-project) using the transformation previously calibrated (Salvadó *et al*, 2019). In the present study, we defined Aβ PET positivity based on the visual read of an expert clinician. Additionally, we defined an "subthreshold Aβ pathology" group that comprise those participants with CL scale between 12 and 30 (Salvadó *et al*, 2019; Milà-Alomà *et al*, 2020).

## Ethics approval and consent to participate

The ALFA+ study (ALFA-FPM-0311) was approved by the Independent Ethics Committee "Parc de Salut Mar", Barcelona, and registered at Clinicaltrials.gov (Identifier: NCT02485730). All participants signed the study's informed consent form that had also been approved by the Independent Ethics Committee "Parc de Salut Mar", Barcelona. The experiments conformed to the principles set out in the WMA Declaration of Helsinki and the Department of Health and Human Services Belmont Report.

## Statistical analyses

We tested for normality of the distribution for each biomarker using the Kolmogorov–Smirnov test and visual inspection of histograms.

### The paper explained

#### Problem

Cerebrospinal fluid (CSF) phosphorylated tau (p-tau) is one of the core biomarkers for Alzheimer's disease (AD). Most p-tau assays target the mid-region (Mid) fragment of the protein and the phosphorylation at threonine-181 (Mid-p-tau181). Recently, new assays have been developed, targeting different phosphorylation sites (T181, T217, T231) or different fragments (N-terminal [N] vs. mid-region tau). Moreover, new p-tau assays in blood have shown high accuracy to detect AD. However, it is less known whether these novel p-tau biomarkers change early in the asymptomatic stages of the Alzheimer's *continuum*, when Aβ pathology is emerging.

#### Results

In this study, we aimed at characterizing the pattern of changes in p-tau in the preclinical stage of the Alzheimer's *continuum*. We performed a head-to-head comparison of the following p-tau biomarkers: (i) CSF Mid-p-tau181 (used as a reference biomarker), (ii) CSF N-p-tau181, (iii) CSF N-p-tau217, (iv) CSF Mid-p-tau231 and (v) plasma N-p-tau181. We found that the novel p-tau biomarkers CSF and plasma N-p-tau181, CSF N-p-tau217 and CSF Mid-p-tau231 increase early in the preclinical stage of the Alzheimer's *continuum*, probably in response to subtle Aβ pathology. Furthermore, plasma N-p-tau181 also significantly increases at this stage, although that increase is milder.

#### Impact

These results suggest that there are early changes in tau metabolism in preclinical Alzheimer, which can be detected with these novel p-tau assays. Therefore, they support the idea of therapeutically targeting tau very early in the disease, as soon as Aβ pathology arises.

The following biomarkers did not follow a normal distribution and were $log_{10}$-transformed: CSF Mid-p-tau181, N-p-tau181, N-p-tau217, Mid-p-tau231, t-tau and NfL, and plasma N-p-tau181 and NfL. The CSF Aβ42/40 ratio followed a normal distribution and was not transformed. We excluded from the main analyses the following extreme values (as defined by the Tukey criteria): 2 CSF N-p-tau181, 1 CSF Mid-p-tau231 and 1 plasma NfL. Including them in the analyses did not change the results.

A *t*-test was used to compare continuous variables between Aβ status groups and Pearson's chi-square test to compare categorical variables. The association between each biomarker and age was assessed with a linear regression adjusting for the effect of sex. These analyses were further conducted stratifying by Aβ group and, additionally, including in the linear regression a "Age × Aβ status" interaction term. We also tested the effect of sex in each biomarker in a one-way ANCOVA adjusted by age.

The comparisons of the p-tau biomarkers between Aβ status groups (as defined by either CSF Aβ42/40 ratio, Aβ PET visual read or Centiloid categories) were performed with a one-way ANCOVA adjusted for the effect of age and sex. Bonferroni-corrected *post hoc* pairwise comparisons were conducted if there were more than two categories. Changes of p-tau biomarkers as a function of CSF Aβ42/40 or Aβ PET CL were assessed with a linear regression adjusting for the effect of age and sex. These analyses were further conducted stratifying by Aβ group and, additionally, including in the linear regression a "Aβ biomarker × Aβ status" interaction term.

We tested the accuracy of p-tau biomarkers to discriminate between A+ and A− groups with a receiver operating characteristic (ROC) analysis. We computed areas under the curve (AUC), the 95% confidence intervals and tested whether they were significantly different from the null hypothesis that the AUC equals 0.50, which corresponds to a random test. We compared the AUCs of two ROC curves using the DeLong test (Robin *et al*, 2011). In Appendix Table S1, we additionally computed the *P*-values applying a false discovery rate (FDR) multiple comparison correction following the Benjamini–Hochberg procedure.

We used a Spearman rank-order correlation to test the correlation between p-tau biomarkers. Comparisons of p-tau biomarkers between AT groups were conducted with a one-way ANCOVA adjusted for the effect of age and sex, followed by Bonferroni-corrected *post hoc* pairwise comparisons. Associations between p-tau biomarkers and CSF NfL were performed with a linear regression adjusted for age and sex. We also performed the analyses stratifying by Aβ group and including the "CSF NfL × Aβ status" interaction term.

We plotted biomarkers levels as a function of CSF Aβ42/40 or Aβ PET as previously described (Milà-Alomà *et al*, 2020). Each biomarker value was corrected by age and sex, and we converted to a z-score using the A−T− group (i.e. normal AD biomarkers) as a reference group. We modelled the relationship between each biomarker and the proxies of disease progression (CSF Aβ42/40 or Aβ PET) using a robust local weighted regression method (rlowess; "smooth" function in Matlab and a span of 300), and we plotted the resulting model (Cleveland, 1979; Orfanidis, 1996). This approach allowed us to define more precisely the biomarker changes in the Alzheimer's *continuum* and therefore overcome the limitation of simplifying the preclinical stage in discrete stages.

All tests were 2-tailed, with a significance level of α = 0.05. Statistical analyses were performed in SPSS IBM 20.0 and R software (http://www.r-project.org/). Figures were built using R and Matlab (v2018b).

# Data availability

This study includes no data deposited in external repositories.

**Expanded View** for this article is available online.

## Acknowledgements

This publication is part of the ALzheimer's and FAmilies (ALFA) study. The authors would like to express their most sincere gratitude to the ALFA project participants and relatives without whom this research would have not been possible. We also thank Tania Menchón, Carolina Herrero, Noemí Carranza, Charlotte Frick, Linnéa Lagerstedt, Irina Nilsson and Cindy François for technical assistance. Collaborators of the ALFA study are Annabella Beteta, Raffaele Cacciaglia, Alba Cañas, Carme Deulofeu, Irene Cumplido, Ruth Dominguez, Maria Emilio, Carles Falcon, Sherezade Fuentes, Laura Hernandez, Gema Huesa, Jordi Huguet, Paula Marne, Tania Menchón, Grégory Operto, Albina Polo, Sandra Pradas, Anna Soteras, Marc Vilanova and Natalia Vilor-Tejedor. The authors thank Roche Diagnostics International Ltd. for providing the kits to measure CSF biomarkers. ELECSYS, COBAS, and COBAS E are registered trademarks of Roche. Authors also thank GE Healthcare for kindly providing the [18F]flutemetamol doses of

ALFA+ study participants. The project leading to these results has received funding from "la Caixa" Foundation (ID 100010434), under agreement LCF/PR/GN17/50300004 and the Alzheimer's Association and an international anonymous charity foundation through the TriBEKa Imaging Platform project (TriBEKa-17-519007). Additional support has been received from the Universities and Research Secretariat, Ministry of Business and Knowledge of the Catalan Government under the grant no. 2017-SGR-892. MS-C received funding from the European Union's Horizon 2020 Research and Innovation Program under the Marie Sklodowska-Curie action grant agreement No 752310 and currently receives funding from Instituto de Salud Carlos III (PI19/00155) and from the Spanish Ministry of Science, Innovation and Universities (Juan de la Cierva Programme grant IJC2018-037478-I). TKK holds a Brightfocus postdoctoral fellowship (#A2020812F) and was further supported by the Swedish Alzheimer Foundation (Alzheimerfonden; #AF-930627), the Swedish Dementia Foundation (Demensförbundet), Gamla Tjänarinnor, the Aina (Ann) Wallströms and Mary-Ann Sjöbloms Foundation, the Gun and Bertil Stohnes foundation, and the Anna Lisa and Brother Björnsson's Foundation. JDG is supported by the Spanish Ministry of Science and Innovation (RYC-2013-13054). OGR is supported by the Spanish Ministry of Science, Innovation and Universities (FJCI-2017-33437). ASV is the recipient of an Instituto de Salud Carlos III Miguel Servet II fellowship (CP II 17/00029). EMAU is supported by the Spanish Ministry of Science, Innovation and Universities—Spanish State Research Agency (RYC2018-026053-I). CM was supported by the Spanish Ministry of Economy and Competitiveness (grant no IEDI-2016-00690). HZ is a Wallenberg Scholar supported by grants from the Swedish Research Council (#2018-02532), the European Research Council (#681712), Swedish State Support for Clinical Research (#ALFGBG-720931), the Alzheimer Drug Discovery Foundation (ADDF), USA (#201809-2016862), and the UK Dementia Research Institute at UCL. KB is supported by the Swedish Research Council (#2017-00915), the Alzheimer Drug Discovery Foundation (ADDF), USA (#RDAPB-201809-2016615), the Swedish Alzheimer Foundation (#AF-742881), Hjärnfonden, Sweden (#FO2017-0243), the Swedish state under the agreement between the Swedish government and the County Councils, the ALF-agreement (#ALFGBG-715986), and European Union Joint Program for Neurodegenerative Disorders (JPND2019-466-236).

## Author contributions

MS-C, TKK, NJA, JLR, HZ and KB conducted the biomarker measurements. MS-C, TKK, NJA, JLR, MM-A, JDG, HZ, KB and JLM analysed and interpreted the data. GK provided the NeuroToolKit and organized samples measurements. ES and EVM provided the Mid-p-tau231 ELISA. MS-C, MM-A, JDG, GS, CM, KF, MS, OG-R, EMA-U, AS-V, GS-B, JMG and JLM contributed with ALFA+ participants data and samples. MS-C, TKK, NJA, HZ and JLM designed the study and wrote the manuscript. All authors critically reviewed and approved the final manuscript.

## Conflict of interest

JDG has given lectures in symposia sponsored by the following for-profit companies: General Electric, Philips and Biogen. GK is a full-time employee of Roche Diagnostics GmbH. ES is an employee and EVM is a co-founder of ADx NeuroSciences. HZ has served at scientific advisory boards for Denali, Roche Diagnostics, Wave, Samumed and CogRx, has given lectures in symposia sponsored by Fujirebio, Alzecure and Biogen, and is a co-founder of Brain Biomarker Solutions in Gothenburg AB (BBS), which is a part of the GU Ventures Incubator Program. KB has served as a consultant or at advisory boards for Abcam, Axon, Biogen, Lilly, MagQu, Novartis and Roche Diagnostics, and is a co-founder of Brain Biomarker Solutions in

Gothenburg AB, a GU Ventures-based platform company at the University of Gothenburg. JLM has served/serves as a consultant or at advisory boards for the following for-profit companies, or has given lectures in symposia sponsored by the following for-profit companies: Roche Diagnostics, Genentech, Novartis, Lundbeck, Oryzon, Biogen, Lilly, Janssen, Green Valley, MSD, Eisai, Alector, BioCross, GE Healthcare, ProMIS Neurosciences. The remaining authors declare that they have no conflict of interest. MSC has given lectures in symposia sponsored by ROCHE DIAGNOSTICS, S.L.U.

## For more information

https://www.barcelonabeta.org/

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
