## [Review Process File · EMBO Molecular Medicine]

Novel tau biomarkers phosphorylated at T181, T217 or T231 rise in the initial stages of the preclinical Alzheimer's continuum when only subtle changes in A β pathology are detected

Marc Suárez-Calvet, Thomas Karikari, Nicholas Ashton, Juan Lantero Rodríguez, Marta Milà-Alomà, Juan Domingo Gispert, Gemma Salvadó, Carolina Minguillón, Karine Fauria, Mahnaz Shekari, Oriol Grau-Rivera, Eider Arenaza-Urquijo, Aleix Sala-Vila, Gonzalo Sánchez-Benavides, José Maria González-de-Echávarri, Gwendlyn Kollmorgen, Erik Stoops, Eugeen Vanmechelen, Henrik Zetterberg, Kaj Blennow, and José Luis Molinuevo

DOI: 10.15252/emmm.202012921

Corresponding author(s): José Luis Molinuevo (jlmolinuevo@barcelonabeta.org) , Kaj Blennow (Kaj.blennow@neuro.gu.se)

Review Timeline:

Submission Date:	11th Jun 20
Editorial Decision:	7th Jul 20
Revision Received:	14th Aug 20
Editorial Decision:	24th Aug 20
Revision Received:	9th Sep 20
Accepted:	9th Sep 20

Editor: Jingyi Hou

Transaction Report:

7th Jul 2020

Dear Dr. Molinuevo,

Thank you for the submission of your manuscript to EMBO Molecular Medicine. We have now received feedback from the three referees whom we asked to evaluate your manuscript. As you will see from the reports below, the referees acknowledge the potential interest of the study. However, they also raise substantial concerns about your work, which should be convincingly addressed in a major revision of the present manuscript.

While the referee #2 and #3 are supportive, referee#1 is concerned about the overall novelty presented in the study in comparison to the existing studies. In particular, during our pre-decision cross-commenting process (in which the referees are given the chance to make additional comments, including on each other's reports), referee #2 added "My understanding is that the novelty of this study is the use of different, more sensitive N-terminal tau antibodies in the biomarker assays and the combination of 5 different tau biomarkers in all study participants, most of whom were cognitively unimpaired but had some early changes in amyloid beta protein suggesting initiation of AD pathology. I agree with Referee #1 that a comparison of the CSF and plasma levels of N-p-tau181 would be helpful. The authors state that a longitudinal study is needed -- which I am guessing is probably underway. In my opinion, the strengths outweigh the limitations/weaknesses of the paper and add to a growing literature about early tau changes in AD pathogenesis."

In light of referee #1's comment on the novelty, attention should be given to placing the findings in the context of previous literature and to highlighting the novelty of the current study. All other issues raised by the referees need to be satisfactorily addressed as well. We would welcome the submission of a revised version within three months for further consideration. Please note that EMBO Molecular Medicine strongly supports a single round of revision and that, as acceptance or rejection of the manuscript will depend on another round of review, your responses should be as complete as possible.

We are aware that many laboratories cannot function at full efficiency during the current COVID-19/SARS-CoV-2 pandemic and have therefore extended our "scooping protection policy" to cover the period required for a full revision to address the experimental issues. Please let me know should you need additional time, and also if you see a paper with related content published elsewhere.

I look forward to receiving your revised manuscript.

Yours sincerely,
Jingyi Hou

Jingyi Hou
Editor
EMBO Molecular Medicine

*** Instructions to submit your revised manuscript ***

** PLEASE NOTE ** As part of the EMBO Publications transparent editorial process initiative (see our Editorial at <https://www.embopress.org/doi/pdf/10.1002/emmm.201000094>), EMBO Molecular Medicine will publish online a Review Process File to accompany accepted manuscripts.

To submit your manuscript, please follow this link:

Link Not Available

- 1) a .docx formatted version of the manuscript text (including Figure legends and tables). Please make sure that the changes are highlighted to be clearly visible to referees and editors alike.
- 2) separate figure files*
- 3) supplemental information as Expanded View and/or Appendix. Please carefully check the authors guidelines for formatting Expanded view and Appendix figures and tables at <https://www.embopress.org/page/journal/17574684/authorguide#expandedview>
- 4) a letter INCLUDING the reviewers' reports and your detailed responses to their comments (as Word file)

Also, and to save some time should your paper be accepted, please read below for additional information regarding some features of our research articles:

- 5) The paper explained: EMBO Molecular Medicine articles are accompanied by a summary of the articles to emphasize the major findings in the paper and their medical implications for the non-

specialist reader. Please provide a draft summary of your article highlighting

6) For more information: There is space at the end of each article to list relevant web links for further consultation by our readers. Could you identify some relevant ones and provide such information as well? Some examples are patient associations, relevant databases, OMIM/proteins/genes links, author's websites, etc...

7) Author contributions: the contribution of every author must be detailed in a separate section (before the acknowledgments).

8) EMBO Molecular Medicine now requires a complete author checklist (<https://www.embopress.org/page/journal/17574684/authorguide>) to be submitted with all revised manuscripts. Please use the checklist as a guideline for the sort of information we need WITHIN the manuscript as well as in the checklist. This is particularly important for animal reporting, antibody dilutions (missing) and exact p-values and n that should be indicated instead of a range.

9) Every published paper now includes a 'Synopsis' to further enhance discoverability. Synopses are displayed on the journal webpage and are freely accessible to all readers. They include a short stand first (maximum of 300 characters, including space) as well as 2-5 one sentence bullet points that summarise the paper. Please write the bullet points to summarise the key NEW findings. They should be designed to be complementary to the abstract - i.e. not repeat the same text. We encourage inclusion of key acronyms and quantitative information (maximum of 30 words / bullet point). Please use the passive voice. Please attach these in a separate file or send them by email, we will incorporate them accordingly.

You are also welcome to suggest a striking image or visual abstract to illustrate your article. If you do please provide a jpeg file 550 px-wide x 400-px high.

10) A Conflict of Interest statement should be provided in the main text

11) Please note that we now mandate that all corresponding authors list an ORCID digital identifier. This takes <90 seconds to complete. We encourage all authors to supply an ORCID identifier, which will be linked to their name for unambiguous name identification.

Currently, our records indicate that there is no ORCID associated with your account.

Please click the link below to provide an ORCID:

Link Not Available

12) The system will prompt you to fill in your funding and payment information. This will allow Wiley to send you a quote for the article processing charge (APC) in case of acceptance. This quote takes into account any reduction or fee waivers that you may be eligible for. Authors do not need to pay any fees before their manuscript is accepted and transferred to our publisher.

Photos 400-800 DPI

*Additional important information regarding figures and illustrations can be found at <http://bit.ly/EMBOPressFigurePreparationGuideline>

***** Reviewer's comments *****

Referee #1 (Comments on Novelty/Model System for Author):

Two recent studies have investigated the relationship of these p-tau fragments in AD, it is unclear what the present study adds to that literature, other than the focus on cognitively unimpaired individuals.

Referee #1 (Remarks for Author):

Summary:

This paper investigates the relationship of p-tau markers for different phosphorylation sites, and mid or N term fragments in CSF, and N-ptau181 in plasma with amyloid changes in cognitively unimpaired older individuals. CSF ptau-181 and ptau-217 have previously been investigated in relation to clinical status, and tau PET across clinical stages in AD. A strength of this study is the large sample of cognitively unimpaired individuals, of whom a large percentage also has amyloid PET. Still, several issues remain unclear:

- The rationale of the study could be more clearly introduced. Now it presents a lot of literature, but the focus and precise rationale for all analyses performed remains unclear. In particular a more clear presentation of recent studies on these CSF ptau markers (e.g., Janelidze 2020 & Barthelmy 2020 studies), what they found and what new aspects will be researched here, and why it is important to study those questions with both CSF and PET amyloid measures. It is explicitly stated in the introduction section: 'However, within the Alzheimer's continuum [undefined], there is a high correlation between CSF p-tau and t-tau and it is difficult to disentangle to what extent these biomarkers reflect tau pathophysiology, AD-triggered neurodegeneration or both' This would be interesting to investigate indeed, but it is not clear how this study will resolve this question, and to what extent the different tau phosphorylation sites can help answer this question, and most importantly, no measures for tau pathology included other than CSF and plasma are included.
- Also, the rationale for comparing plasma p-tau in with plasma NFL to predict amyloid status is not clear. Correlating and comparing between plasma p-tau and CSF p-tau would be more

straightforward in terms of validation, also given the notion that NFL is increased in many other neurodegenerative disorders than AD?

-Although previous research has shown that CSF abeta can be abnormal before amyloid PET, the discordance between CSF and PET amyloid status in the present study is very large: only 39% of individuals with abnormal CSF have an abnormal PET, compared to e.g., 63% in ADNI and 76% in the Biofinder cohorts as reported in Palmqvist et al., (2017) in participants with normal cognition. This suggests that the ab42/40 cutoff used in the present paper may not reflect amyloid pathology. This is further supported by the notion that p-tau predictive values are systematically higher for prediction of amyloid status based PET versus CSF. How can these differences be explained? (also note that the text at p12 implies that predictive accuracy for ptau markers is best for CSF and PET CL12, but this does not match the results presented in table 3, which shows highest absolute number for PET visual read).

The A- group showed a negative relationship with most p-tau measures and CSF ab42/40, while this was positive for the PET CL scale. To dissociate sub-threshold changes from physiological changes it would be of interest to show how CSF ab42/40 and PET CL correlate with each other in this group, and for A- classified based on PET only to see how CSF A+ may influence those results.

-PET amyloid subthreshold effects are explored by comparing subgroups on ptau measures. Most subjects, however, have a CL value < 12: Do the relationships between ptau measures and CL in this group still hold? That would imply that 'subthreshold' may span a larger range of values. Also rerunning analyses with interaction term based on amyloid PET normal/abnormal for the 12 CL threshold would further formally test differences in these relationships between normal and abnormal groups.

Where do the individuals with >12CL fall in the CSF ab42/40 measures? And conversely, where do the CSF ab42/40 abnormal individuals fall in the PET CL plots? Could these be color coded in the regression scatterplots of figures 2 and 3?

-From p13 onward p-tau changes across the preclinical AD stages are modelled cross-sectionally, by taking ab42/ab40 as a proxy for disease progression. However, there is no clear argumentation provided whether this is a valid approach, and this seems to conflict with statements in the introduction section that p-tau measures are more closely related to cognitive decline. It is not clear what these analyses add over the previous more straightforward analyses.

Other points:

-The introduction section is long and unclear in terms of the rationale for the specific analyses that are performed. There are already CSF papers out on the different phosphor-tau's, it should be better explained what the added value of the present analyses is.

-In Table 3 the A+ and A- numbers seem to be swapped.

Referee #2 (Comments on Novelty/Model System for Author):

Examination of novel, highly sensitive tau biomarkers in CSF and plasma of individuals in the preclinical stage of AD is timely and relevant for early diagnosis, tracking disease progression, monitoring drug engagement and therapeutic outcomes.

Referee #2 (Remarks for Author):

Suarez-Calvet and colleagues from the ALFA study program report new and exciting data indicating that several novel p-tau biomarkers including CSF N-p-tau181, CSF N-p-tau217, CSF Mid-p-tau231 and to a lesser degree, plasma N-p-tau181 increase at very early stages of the AD continuum, when only subtle changes in A β are detected (between 12 and 30 CL) and individuals

are cognitively unimpaired. The 381 study participants were part of a well-characterized cohort of CU individuals and included a large number of middle-aged (45-75 yo), preclinical AD determined by CSF A β ratio (of which 76% were A+ T-). The novel CSF biomarkers increased early in preclinical AD, increased with A β pathology and were capable of distinguishing A+ from A- individuals. The novel biomarkers were shown to be more sensitive than the commonly used CSF Mid-p-tau181 or total tau biomarkers. In addition, CSF N-p-tau217 was increased in A+ T- early stage preclinical AD but not in A- T+ non-AD participants, suggesting that it may be useful to distinguish AD from non-AD tauopathies at early stages of pathogenesis. CSF Mid-p-tau231 had the biggest increase and has been detected in pre-neurofibrillary tangles (NFT) before filaments suggesting that it is an early change that occurs before NFT formation. The plasma N-p-tau181 biomarker increase was less robust than the novel CSF tau biomarkers but was higher than the increase in plasma neurofilament light chain (NfL). The authors were careful to note the limitations of this cross-sectional study, including the absence of tau PET data, and acknowledged that longitudinal studies are needed. The paper is extremely well-written, clear and straightforward. The use of such novel, sensitive CSF biomarkers will help inform future clinical studies in the preclinical AD population and may be useful for measuring target engagement in tau therapeutic trials and monitoring tau in anti-amyloid trials.

Minor Comments:

1. Results - Section "Novel p-tau biomarkers detect amyloid- β pathology more accurately in cognitively unimpaired individuals": the last sentence appears to have an error. It reads "For plasma N-p-tau181, however, the AUC to discriminate between A β and A β - in CU individuals was lower (Table 3; Fig 4B, D, F) and similar to that of plasma N-p-tau181, plotted as a comparison."
2. Do the plasma N-p-tau181 levels correlate with the CSF N-p-tau181 levels?
3. The authors mention in the Discussion that "CSF N-p-tau217 was slightly higher in women than men". Should this be in the Results section? Is this the only novel biomarker that showed sex-specific effects (even if only subtle)?
4. Did the presence of an APOE ϵ 4 allele correlate with higher CSF and/or plasma tau biomarker levels?
5. Do the authors know if any of these novel tau biomarkers can be detected in urine?

Referee #3 (Remarks for Author):

Suarez-Calvert, Karikari and coworkers investigate a set of novel p-tau assays as potential biomarkers for AD. The authors find that in N-terminal-inclusive pT181 and pT217 assays, evidence of p-Tau increases in CSF and plasma are evident in early stages of preclinical AD. The authors note the limitations that this is a cross-sectional study and a mix of analytical platforms may to some degree confound some of the conclusions. Nonetheless, this study can stand alone as an interesting set of observations that could add important insight into how core biomarkers fit into the AD continuum. I have only one comment in regard to additional analysis. Did the authors look at APOE4 carriers in the Ab- group to see if p-tau levels were significantly different? It is somewhat predictable that APOE4 carriers might have very subtle levels of Ab deposition in this group that might lead to increases in p-tau levels for N-term 217 or 181 in CSF.

Please find enclosed the revised version of our manuscript (EMM-2020-12921), entitled “*Novel tau biomarkers phosphorylated at T181, T217 or T231 rise in the initial stages of the preclinical Alzheimer’s continuum when only subtle changes in A β pathology are detected*”. We would like to thank you for giving us the opportunity to submit a revised version of the manuscript. As you kindly suggested, we have emphasized the novelty of our study and we have placed our findings in the context of recently published literature. We also performed the comparison between CSF and plasma N-p-tau181 (see new Table 3 and Figure 4), as you indicated.

We would like to take this opportunity to thank the reviewers for their support and interesting suggestions, which has certainly improved the quality of this article. A point-by-point response follows. The reviewer comments are quoted and in bold, and our responses are in plain text.

Reviewer #1:

1. **“The rationale of the study could be more clearly introduced. Now it presents a lot of literature, but the focus and precise rationale for all analyses performed remains unclear. In particular a more clear presentation of recent studies on these CSF ptau markers (e.g., Janelidze 2020 & Barthelemy 2020 studies), what they found and what new aspects will be researched here, and why it is important to study those questions with both CSF and PET amyloid measures. It is explicitly stated in the introduction section: 'However, within the Alzheimer's continuum [undefined], there is a high correlation between CSF p-tau and t-tau and it is difficult to disentangle to what extent these biomarkers reflect tau pathophysiology, AD-triggered neurodegeneration or both' This would be interesting to investigate indeed, but it is not clear how this study will resolve this question, and to what extent the different tau phosphorylation sites can help answer this question, and most importantly, no measures for tau pathology included other than CSF and plasma are included.”**

We agree with the reviewer that clarification of the rationale of this study is required more explicitly in the main text and put it in the context of recent literature. In response, we have done major changes in the introduction section.

We have clarified the rationale of the study and highlighted its novelty:

Introduction, page 5-6: “Despite the recent breakthrough developments in understanding p-tau as both a CSF and blood biomarker, it is less known whether specific p-tau biomarkers change early in the continuum of sporadic Alzheimer, when only subtle, incipient changes in A β pathology are present. Studying these early stages of the disease is particularly relevant for the p-tau biomarkers since their changes probably precedes NFT pathology (Barthélemy et al, 2020c). Amid the new p-tau assays being developed (targeting different phosphorylation sites, targeting N-terminal vs mid region tau or using different platforms), it is important to perform a head-to-head comparison of these assays. Moreover, there are promising tau phosphorylations, such p-tau231, that have not been yet investigated in asymptomatic stages.”

We have stressed the importance of the more recent studies on p-tau, especially those related to p-tau217:

Introduction, page 5: *“Besides p-tau181, recent strong data indicates that p-tau217, measured both in CSF and in plasma, accurately predicts A β pathology in both symptomatic and asymptomatic stages and is an excellent biomarker to discriminate AD from healthy controls and other neurodegenerative diseases (Barthelemy et al., 2015; Barthélemy, Bateman, et al., 2020; Barthélemy et al., 2017; Barthélemy, Li, et al., 2020; Janelidze et al., 2020). In autosomal-dominant AD, CSF p-tau217 even increases two decades before tau PET (Barthélemy, Li, et al., 2020).”*

In the discussion, we have also discussed the new plasma p-tau217 studies and we have acknowledged one of the limitations of our study, that is not including plasma p-tau217:

Discussion, page 17: *“This probable higher specificity is in line with recent studies demonstrating that plasma p-tau217 detects AD pathology with a very high accuracy (Barthélemy, Horie, Sato, & Bateman, 2020; Palmqvist et al., 2020).”*

Discussion, page 22: *“Finally, our study did not include plasma p-tau217. While our manuscript was being reviewed, two studies were published showing the high value of this biomarker to identify AD (Barthélemy, Horie, et al., 2020; Palmqvist et al., 2020).”*

We highlighted the importance of including A β CSF and PET in the same study:

Introduction, page 6: *“In order to ensure robustness, we used both CSF and PET as biomarkers of A β pathology, which reflect different aspects of A β pathology (i.e., soluble A β and fibrillar A β aggregates, respectively).”*

Finally, as suggested by the reviewer, we have shortened the introduction and we have deleted an entire paragraph that do not clearly relate to the aims of the study.

- 2. “Also, the rationale for comparing plasma p-tau in with plasma NFL to predict amyloid status is not clear. Correlating and comparing between plasma p-tau and CSF p-tau would be more straightforward in terms of validation, also given the notion that NFL is increased in many other neurodegenerative disorders than AD?”**

We understand the reviewers’ point and we concur. In the revised manuscript, we directly compare CSF and plasma N-p-tau181 and we included the following new sentence in the main text:

Results, page 12: *“For plasma N-p-tau181, however, the AUCs to discriminate between A+ and A- in CU individuals were significantly lower (Table 3; Fig 4B) than its CSF counterpart, CSF N-p-tau181. Yet, plasma N-p-tau181 had and similar AUCs to those of plasma NfL (except for A β PET CL12 where plasma N-p-tau181 had a significantly higher AUC), plotted as a comparison.”*

Moreover, we have merged the CSF and plasma ROC figures (see new Figure 4 below) so that all the ROC curves can be compared in the same graph. Also, we have conducted a comparison of all the biomarkers AUCs with a DeLong test, without separating between CSF and plasma biomarkers (see new Table 3).

The correlation between CSF and plasma N-p-tau181 is shown in Fig EV3 and in the following sentence in the main text:

Results, page 12: *“the correlation between CSF and plasma N-p-tau181 was weaker but still significant (Fig EV3). Of note, the correlation of CSF and plasma N-p-tau181 was higher in the A+ ($r_s = +0.353$, $P < 0.0001$) than in the A- group ($r_s = +0.191$, $P = 0.003$)”*

New Figure 4. Discrimination of cognitively unimpaired A β -positive from A β negative individuals by p-tau biomarkers.

ROC analysis was performed to test the accuracy to discriminate between A β -positive from A β -negative individuals. A β positivity was defined as CSF A β 42/40 < 0.071 (A), A β PET visual read (B) or A β PET Centiloid (CL) > 12 (C).

Abbreviations: CSF, cerebrospinal fluid; Mid, mid-region; NfL, neurofilament light; N, N-terminal; p-tau, phosphorylated tau; t-tau, total tau.

3. **“Although previous research has shown that CSF abeta can be abnormal before amyloid PET, the discordance between CSF and PET amyloid status in the present study is very large: only 39% of individuals with abnormal CSF have an abnormal PET, compared to e.g., 63% in ADNI and 76% in the Biofinder cohorts as reported in Palmqvist et al., (2017) in participants with normal cognition. This suggest that the ab42/40 cutoff used in the present paper may not reflect amyloid pathology. This is further supported by the notion that p-tau predictive values are systematically higher for prediction of amyloid status based PET versus CSF. How can these differences be explained? (also note that the text at p12 implies that predictive accuracy for ptau markers is best for CSF and PET CL12, but this does not match the results presented in table 3, which shows highest absolute number for PET visual read).”**

The reviewer raises a very important issue. The reviewer is correct in pointing out that the cohort studied (ALFA+) has a higher number of A β CSF and PET mismatches compared to other studies. Regarding this, we would like to emphasise the following points:

- a. The ALFA+ cohort, which is enriched by individuals at risk of AD, has a high percentage of individuals that are cognitively unimpaired and are A β -positive but Tau-negative (A+T-), using as biomarkers CSF A β 42/40 and CSF M-p-tau181. In other words, there are a number of individuals in the earliest stage of the Alzheimer's *continuum*, that is in the Alzheimer's pathological change stage (as defined by the 2018 NIA-AA Research Framework). Therefore, it is not surprising we found a higher number of CSF/PET mismatches, because there may be individuals that already have soluble A β changes (as shown by CSF) but not A β fibrillar changes (as shown by PET). This is in fact one of the main strengths of this cohort.
- b. The CSF A β 42/40 ratio cutoff derived herein is not intended to be used for diagnostic purposes and we acknowledge that it might not be generalizable to clinical cohorts. However, our goal is to derive sensitive cutoffs that capture the very early stages of the disease *continuum*, when A β pathophysiological changes are starting. The typical cutoffs derived in symptomatic AD (e.g. based on a comparison against A β PET or seeking an optimal discrimination between a control and a pathological group) are very useful in clinical population but they might not detect very early preclinical Alzheimer's. This is why we applied a data-driven approach, namely a Gaussian Mixture Modelling (GMM), which allows to describe a normal range of CSF A β 42/40 ratios in the non-pathological group and derive a cutoff defined as the mean minus 2 standard deviations of this Gaussian distribution. In fact, GMM has already been used to derive cohort-specific CSF A β cutoffs, showing a robust ability to discriminate normal and pathologic distributions of A β 42 (Bertens, Tijms, Scheltens, Teunissen, & Visser, 2017; De Meyer et al., 2010; Willemsse et al., 2018). The cutoff used in the ALFA+ cohort has already been published (Milà - Alomà et al., 2020) Milà-Alomà et al, JPAD, in press).

Considering these points, it is reasonable that the p-tau biomarkers predictive values are better for prediction of PET than CSF A β status. CSF-based A β status captures an early stage of the disease than PET-based A β status.

We would also like to draw the reviewer's attention to new Figure 3 (see below), where we coloured the CSF-based A β -status in the graph with A β PET Centiloids. Please note that most CSF-based A β -positive (A+) individuals (coloured in Turquoise) are just below the 12CL threshold, which agrees with the idea that CSF cutoffs detects even more subtle A β changes than A β PET.

Finally, the reviewer is right that that the sentence is page 12 about the predictive accuracy of p-tau biomarkers in not correct and misleading and we have removed it from the revised manuscript.

New Figure 3. Effect of A β pathology (A β PET) on p-tau biomarkers

(A, D, G, J, M) Dot and box-plot showing comparison between each of the p-tau biomarker between the A β negative (A-; blue) and the A β positive (A+; red) groups. A β positivity was defined with A β PET visual read. P-values were assessed by a one-way ANCOVA adjusted by age and sex.

(B, E, H, K, N) Scatter plots depicting the changes between each p-tau biomarker as a function of A β PET Centiloids (CL). The standardized regression coefficients (β) and the P -values were computed using a linear model adjusting for age and sex. The solid lines indicate the regression line and the 95% confidence intervals. The dashed green lines indicate the CL12 and CL30 cutoffs. Participants were also colour-coded based on the CSF A β 42/40 ratio (A-, black; A+, turquoise).

(C, F, I, L, O) Dot and box-plot depicting comparison between each of the p-tau biomarker between Centiloid scale groups. The group between 12 and 30 CL (grey) indicates the subtle A β pathology group. P -values were assessed by a one-way ANCOVA adjusted by age and sex, followed by a Bonferroni-corrected *post hoc* pairwise comparison. Abbreviations: CSF, cerebrospinal fluid; Mid, mid-region; N, N-terminal; p-tau, phosphorylated tau.

- 4. “The A- group showed a negative relationship with most p-tau measures and CSF ab42/40, while this was positive for the PET CL scale. To dissociate sub-threshold changes from physiological changes it would be of interest to show how CSF ab42/40 and PET CL correlate with each other in this group, and for A- classified based on PET only to see how CSF A+ may influence those results.”**

Following reviewer suggestion, we have tested the correlation between CSF A β 42/40 and A β PET CL in the A- group (Figure for the reviewers 1). Of note, CSF A β 42/40 and A β PET CL did not correlate in the A- group (Spearman's correlation coefficient, $r_s = -0.049$, $P = 0.472$), while there was a significant correlation in the A+ group ($r_s = -0.489$, $P < 0.0001$).

Figure for the reviewers 1. Correlation between A β PET and CSF A β 42/40 stratified by A β -status (as defined by the A β 42/04 ratio).

Regarding the second question, there were 73 (25.4%) A+ individuals (based on the CSF A β 42/40 criteria) among the 287 individuals that were classified as A- based on the A β PET visual read. If we remove these CSF A+ individuals from the analyses, the associations between the p-tau biomarkers and A β PET CL in the PET A- group are not significant anymore (Table for the reviewer 1).

Table for the reviewers 1. Association of p-tau biomarkers and A β PET CL scale in A- individuals (as defined by A β PET visual read). We show the results removing the 73 individuals that are A+ based on the CSF A β 42/40 ratio and including them (as in the main text; show here in gray for comparison).

	A- (A β PET visual read)					
	Removing CSF A β 42/40 A+ (n = 214)			Not removing CSF A β 42/40 A+ (n = 287)		
	β (SE)	P	eta ²	β (SE)	P	eta ²
CSF Mid-p-tau181	+0.002 (0.069)	0.981	0.000	+0.074 (0.059)	0.210	0.006
CSF N-p-tau181	+0.015 (0.068)	0.828	0.0003	+0.231 (0.058)	<0.0001*	0.054
CSF N-p-tau217	+0.029 (0.069)	0.676	0.0009	+0.266 (0.058)	<0.0001*	0.072
CSF Mid-p-tau231	+0.046 (0.069)	0.502	0.002	+0.271 (0.057)	<0.0001*	0.073
Plasma N-p-tau181	+0.103 (0.069)	0.137	0.011	+0.217 (0.058)	0.0002*	0.048

This later result suggests that the significant associations between p-tau biomarkers and A β PET Centiloids observed in the A- (as defined by A β PET visual read) are probably driven by those that already have a decreased CSF A β 42/40 ratio. In figure 3, we have colour-code the individuals based on the CSF A β 42/40 ratio.

We thank the reviewer for the interesting observation. Although there maybe physiological changes in p-tau biomarkers in healthy individuals, these results suggest that they change as soon as there are changes in A β metabolism, even with low A β centiloids in A β PET.

5. **“PET amyloid subthreshold effects are explored by comparing subgroups on ptau measures. Most subjects, however, have a CL value < 12: Do the relationships between ptau measures and CL in this group still hold? That would imply that 'subthreshold' may span a larger range of values. Also rerunning analyses with interaction term based on amyloid PET normal/abnormal for the 12 CL threshold would further formally test differences in these relationships between normal and abnormal groups. Where do the individuals with >12CL fall in the CSF ab42/40 measures? And conversely, where do the CSF ab42/40 abnormal individuals fall in the PET CL plots? Could these be color coded in the regression scatterplots of figures 2 and 3?”**

The reviewer is right that there is a higher number of individuals that are A β PET CL<12. Following reviewer suggestion, we have tested the associations between the p-tau biomarkers and A β PET CL in the CL <12 group and we also run interaction analyses including the term ‘A β PET CL x A β status (CL12)’. These analyses have been included in the new expanded view Table EV2 and it is also shown below.

The results using CL12 to classify A+ and A- are similar to those using the A β PET visual read (in Table 2 in the main text). In both analyses, there is a significant increase of all p-tau biomarkers (except for Mid-p-tau181) as a function of A β PET CL in the A- group. The interaction analyses also render a similar result, except for the fact that the interaction ‘A β PET CL x A β status (CL12)’ for Mid-p-tau231 becomes significant. As the reviewer points out, these may imply that there may be A β subtle changes even below A β PET CL12.

Table EV2. Association of p-tau biomarkers and A β PET CL scale stratifying by A β status (as defined by A β PET CL12 cutoff).

	A β PET CL12 (n = 331)						
	A- (n = 278)			A+ (n = 53)			'A β PET x A β status' interaction
	β (SE)	P	eta ²	β (SE)	P	eta ²	P
CSF Mid-p-tau181	+0.058 (0.060)	0.339	0.003	+0.483 (0.118)	0.0002*	0.254	0.009*
CSF N-p-tau181	+0.115 (0.060)	0.054*	0.014	+0.628 (0.132)	<0.0001*	0.324	0.004*
CSF N-p-tau217	+0.176 (0.060)	0.003*	0.031	+0.610 (0.110)	<0.0001*	0.391	0.246
CSF Mid-p-tau231	+0.165 (0.059)	0.006*	0.027	+0.606 (0.119)	<0.001*	0.350	0.040*
Plasma N-p-tau181	+0.130 (0.060)	0.032*	0.017	+0.253 (0.156)	0.112	0.053	0.859

For each p-tau biomarker we computed the linear regression standardized coefficients (β) and standard errors (SE) as a function of A β PET Centiloid scale. The analyses were performed after stratifying for A β -negative (A-) and A β -positive (A+) groups, as defined by A β PET CL12. We report the eta-squared (eta²) as a measure of the effect size. We also computed the *P*-value for the interaction term 'A β biomarker x A β status'. All analyses were adjusted by age and sex. *Significant differences.

Abbreviations: CL, Centiloid; CSF, cerebrospinal fluid; Mid, mid-region; N, N-terminal; PET, positron emission tomography; p-tau, phosphorylated tau.

In the revised manuscript, we have included Table EV2 and the following sentence in the main text:

Results, page 10: "We repeated the same analyses after defining the A+ group as having a CL > 12 in A β PET, instead of visual read, and the results were similar (Table EV2)."

In the Table for reviewer 2, we answer the next reviewer's question. Among the individuals that were A β PET CL>12, only 1 individual (1.9%) was A- and 52 (98%) were A+ using the CSF A β 42/40 ratio. Among the individuals that were A+ with the CSF A β 42/40 ratio, 62 (54%) were A β PET CL<12 and 52 (46%) are A β PET CL \leq 12. Following reviewer suggestion, we have colour code these individuals in the new Figure 2 (see below) and Figure 3 (see above). Together, these results are consistent with the fact that the CSF A β 42/40 ratio changes earlier in the disease *continuum* than the A β PET.

Table for the reviewers 2. Table with the A β -classification agreement between CSF A β 42/40 and A β PET (as defined by CL12).

		A β PET CL12		
		A-	A+	Total
CSF A β 42/40	A-	216	1	217
	A+	62	52	114
	Total	278	53	331

New Figure 2. Effect of A β pathology (CSF A β 42/40) on p-tau biomarkers

(A, C, E, G, I) Dot and box-plot comparing each of the p-tau biomarker between the A β -negative (A-; blue) and the A β -positive (A+; red) groups. A β positivity was defined as CSF A β 42/40 ratio < 0.071 . The box-

plot depict the median (horizontal bar), interquartile range (IQR, hinges) and 1.5 x IQR (whiskers). *P*-values were assessed by a one-way ANCOVA adjusted for age and sex.

(B, D, F, H, J) Scatter plots depicting the changes between each p-tau biomarker as a function of CSF A β 42/40. The horizontal-axes directions were inverted; lower CSF A β 42/40 ratio reflects higher A β pathology. For each A β status group, we computed the standardized regression coefficients (β) and the *P*-values, adjusted for age and sex. The solid lines indicate the regression line and the 95% confidence intervals for each of the A β status groups. The dashed green lines indicate the CSF A β 42/40 cutoff. Participants were also colour-coded based on the A β PET CL scale (≤ 12 CL, black; $>A+$, turquoise; A β PET non-available, grey).

Abbreviations: CSF, cerebrospinal fluid; Mid, mid-region; N, N-terminal; n.a. non-available; p-tau, phosphorylated tau.

6. **“From p13 onward p-tau changes across the preclinical AD stages are modelled cross-sectionally, by taking ab42/ab40 as a proxy for disease progression. However, there is no clear argumentation provided whether this is a valid approach, and this seems to conflict with statements in the introduction section that p-tau measures are more closely related to cognitive decline. It is not clear what these analyses add over the previous more straightforward analyses.”**

We understand the reviewer point and we have better explained why we used this approach:

Methods, page 30-31: *“This approach allowed us to define more precisely the biomarker changes in the Alzheimer’s continuum and therefore overcome the limitation of simplifying the preclinical stage in discrete stages.”*

Studying the long preclinical stage of sporadic AD is difficult because we do not have the concept of “expected years to symptom onset (EYO)” as in autosomal-dominant AD. Also, we cannot use cognitive decline as we are in presymptomatic stages. Our results are consistent using different approaches, which we believe reinforce their robustness.

7. **“Other points:
-The introduction section is long and unclear in terms of the rationale for the specific analyses that are performed. There are already CSF papers out on the different phosphor- tau's, it should be better explained what the added value of the present analyses is.”**

We agree with the reviewer and we have rewritten the introduction section accordingly.

“-In Table 3 the A+ and A- numbers seem to be swapped.”

Thanks for catching the error, we corrected the Table.

Reviewer #2:

“Examination of novel, highly sensitive tau biomarkers in CSF and plasma of individuals in the preclinical stage of AD is timely and relevant for early diagnosis, tracking disease progression, monitoring drug engagement and therapeutic outcomes.”

We appreciate the strong support of the reviewer. They also share some minor comments:

1. **“Results - Section “Novel p-tau biomarkers detect amyloid- β pathology more accurately in cognitively unimpaired individuals”: the last sentence appears to have an error. It reads “For plasma N-p-tau181, however, the AUC to discriminate between A β + and A β - in CU individuals was lower (Table 3; Fig 4B, D, F) and similar to that of plasma N-p-tau181, plotted as a comparison”**

We thank the reviewer for catching the errors. We corrected them in the revised version as follows:

Results, page 11: *“plasma N-p-tau181 had and similar AUCs to those of plasma NfL (except for A β PET CL12 where plasma N-p-tau181 had a significantly higher AUC), plotted as a comparison.”*

2. “Do the plasma N-p-tau181 levels correlate with the CSF N-p-tau181 levels?”

Yes, they do, as shown in Fig EV3. In the revised manuscript we have included the statistics in the main text:

Results, page 12: *“the correlation between CSF and plasma N-p-tau181 was weaker but still significant (Fig EV3). Of note, the correlation of CSF and plasma N-p-tau181 was higher in the A+ ($r_s = +0.353$, $P < 0.0001$) than in the A- group ($r_s = +0.191$, $P = 0.003$)”*

3. “The authors mention in the Discussion that “CSF N-p-tau217 was slightly higher in women than men”. Should this be in the Results section? Is this the only novel biomarker that showed sex-specific effects (even if only subtle)?”

We thank the reviewer to highlight these results. We only found sex differences in CSF N-p-tau217, but not in the other p-tau biomarkers. We do believe that sex and gender differences in biomarkers should be a priority in our research field. This result is explained in the results section as follows:

Results, page 9: *“Interestingly, after adjusting for age, levels of CSF N-p-tau217 were slightly, but significantly, higher in women ($M = 6.71$ pg/ml, $SD = 7.01$) than men ($M = 5.63$, $SD = 5.31$; $P = 0.016$). This difference remained significant after additionally adjusting for A β status ($P = 0.005$). The rest of the CSF or plasma p-tau biomarkers were not affected by sex.”*

4. “Did the presence of an APOE $\epsilon 4$ allele correlate with higher CSF and/or plasma tau biomarker levels?”

This is an interesting point. We conducted a linear regression with each p-tau biomarker as a dependent variable and APOE- $\epsilon 4$ status, age and sex as independent variables. APOE- $\epsilon 4$ status had a significant effect on CSF N-p-tau181 ($P = 0.0001$), CSF N-p-tau217 ($P = 0.00008$), CSF Mid-p-tau231 ($P = 0.019$), but not CSF M-p-tau181 ($P = 0.624$) or plasma N-p-tau181 ($P = 0.224$). However, when we adjusted by amyloid- β status, all the significant results disappeared. This suggests that the effect of APOE- $\epsilon 4$ on p-tau biomarkers is probably related to A β pathology.

5. “Do the authors know if any of these novel tau biomarkers can be detected in urine?”

We are not aware that these p-tau biomarkers can also be measured in urine. We concur with the reviewer that testing biomarkers in matrix different from blood and CSF, such as urine or saliva, would be highly interesting and we will conduct these measurements in future studies.

Reviewer #3:

1. “Suarez-Calvert, Karikari and coworkers investigate a set of novel p-tau assays as potential biomarkers for AD. The authors find that in N-terminal-inclusive pT181 and pT217 assays, evidence of p-Tau increases in CSF and plasma are evident in early stages of preclinical AD. The authors note the limitations that this is a cross-sectional study and a mix of analytical platforms may to some degree confound some of the conclusions. Nonetheless, this study can stand alone as an interesting set of observations that could add important insight into how core biomarkers fit into the AD continuum. I have only one comment in regard to additional analysis. Did the authors look at APOE4 carriers in the A- group to see if p-tau levels were significantly different? It is somewhat predictable that APOE4 carriers might have very subtle levels of Ab deposition in this group that might lead to increases in p-tau levels for N-term 217 or 181 in CSF.”

We thank the reviewer for their support and for the interesting question. We compared the levels of the p-tau biomarkers between the A- (stratifying between *APOE*- ϵ 4 non-carriers and carriers) and the A+ (including both *APOE*- ϵ 4 carriers and non-carriers). The levels of the p-tau biomarkers in the A- and *APOE*- ϵ 4 carriers' group did not differ from those of the A- and *APOE*- ϵ 4 non-carriers group (see figure for the reviewer 2).

Figure for the reviewers 2. Effect of A β pathology (CSF A β 42/40) and *APOE* on p-tau biomarkers. We conducted an ANCOVA adjusted by age and sex followed by a Bonferroni-corrected *post hoc* pairwise comparison.

Taken together, we believe we have addressed all the reviewers' comments and hope that you find our manuscript now acceptable for publication in EMBO Molecular Medicine. Many thanks for considering our manuscript.

Sincerely,

José Luis Molinuevo
Kaj Blennow

24th Aug 2020

Dear Dr. Molinuevo,

Thank you for the submission of your revised manuscript to EMBO Molecular Medicine. We have now received the enclosed report from the referee who was asked to re-assess it. As you will see the referee is now supportive and I am pleased to inform you that we will be able to accept your manuscript pending the following amendments:

1. In the main manuscript file, please do the following:

- Reduce the number of keywords to 5
- Add callouts for Fig 1 A-E (currently only Figure 1 in general is called for). Same for Fig. 4 A & C, Fig. 5A,B,C,D,E.
- remove the yellow color font.
- in legends, provide exact n= and exact p= values, not a range, along with the statistical test used. Some people found that to keep the figures clear, providing an Appendix supplemental table with all exact p-values was preferable. You are welcome to do this if you want to.
- in Material & Methods (and in checklist), include a statement that the experiments conformed to the principles set out in the WMA Declaration of Helsinki and the Department of Health and Human Services Belmont Report.
- in Material & Methods, provide the antibody dilutions that were used for each antibody.

2. Table EV1- EV3 need to be uploaded as separate files (.txt, .xls., etc.) using the file type Expanded View File.

3. Please add a "Data availability" section (placed after the Materials & Methods section). Since there is no large-scale dataset generated in this study, please include the following sentence in this section- "This study includes no data deposited in external repositories".

4. For more information: There is space at the end of each article to list relevant web links for further consultation by our readers. Could you identify some relevant ones and provide such information as well? Some examples are patient associations, relevant databases, OMIM/proteins/genes links, author's websites, etc...

5. We would encourage you to include the source data for figure panels that show essential quantitative information. Additional information on source data and instruction on how to label the files are available at < <https://www.embopress.org/page/journal/17574684/authorguide#sourcedata> >.

6. Our data editor has made a couple of suggestions on your manuscript (see attached), please address.

7. As part of the EMBO Publications transparent editorial process initiative (see our Editorial at <http://embomolmed.embopress.org/content/2/9/329>), EMBO Molecular Medicine will publish online a Review Process File (RPF) to accompany accepted manuscripts.

a. In the event of acceptance, this file will be published in conjunction with your paper and will include the anonymous referee reports, your point-by-point response and all pertinent

correspondence relating to the manuscript. Let us know if you do not agree with this.

8. Please provide a visual abstract (550px width and max 400px height, jpeg format) to illustrate your article on our homepage.

I look forward to seeing a revised version of your manuscript as soon as possible.

Sincerely,

Jingyi

Jingyi Hou
Editor
EMBO Molecular Medicine

*** Instructions to submit your revised manuscript ***

To submit your manuscript, please follow this link:

Link Not Available

1) a .docx formatted version of the manuscript text (including Figure legends and tables)

2) Separate figure files*

3) supplemental information as Expanded View and/or Appendix. Please carefully check the authors guidelines for formatting Expanded view and Appendix figures and tables at <https://www.embopress.org/page/journal/17574684/authorguide#expandedview>

4) a letter INCLUDING the reviewer's reports and your detailed responses to their comments (as Word file).

5) The paper explained: EMBO Molecular Medicine articles are accompanied by a summary of the articles to emphasize the major findings in the paper and their medical implications for the non-specialist reader. Please provide a draft summary of your article highlighting

6) For more information: There is space at the end of each article to list relevant web links for further consultation by our readers. Could you identify some relevant ones and provide such information as well? Some examples are patient associations, relevant databases, OMIM/proteins/genes links, author's websites, etc...

7) Author contributions: the contribution of every author must be detailed in a separate section.

8) EMBO Molecular Medicine now requires a complete author checklist (<https://www.embopress.org/page/journal/17574684/authorguide>) to be submitted with all revised manuscripts. Please use the checklist as guideline for the sort of information we need WITHIN the manuscript. The checklist should only be filled with page numbers where the information can be found. This is particularly important for animal reporting, antibody dilutions (missing) and exact values and n that should be indicated instead of a range.

9) Every published paper now includes a 'Synopsis' to further enhance discoverability. Synopses are displayed on the journal webpage and are freely accessible to all readers. They include a short stand first (maximum of 300 characters, including space) as well as 2-5 one sentence bullet points that summarise the paper. Please write the bullet points to summarise the key NEW findings. They should be designed to be complementary to the abstract - i.e. not repeat the same text. We encourage inclusion of key acronyms and quantitative information (maximum of 30 words / bullet point). Please use the passive voice. Please attach these in a separate file or send them by email, we will incorporate them accordingly.

You are also welcome to suggest a striking image or visual abstract to illustrate your article. If you do please provide a jpeg file 550 px-wide x 400-px high.

10) A Conflict of Interest statement should be provided in the main text

11) Please note that we now mandate that all corresponding authors list an ORCID digital identifier. This takes <90 seconds to complete. We encourage all authors to supply an ORCID identifier, which will be linked to their name for unambiguous name identification.

Currently, our records indicate that the ORCID for your account is 0000-0003-0485-6001.

Please click the link below to modify this ORCID:
Link Not Available

12) The system will prompt you to fill in your funding and payment information. This will allow Wiley to send you a quote for the article processing charge (APC) in case of acceptance. This quote takes into account any reduction or fee waivers that you may be eligible for. Authors do not need to

pay any fees before their manuscript is accepted and transferred to our publisher.

Photos 400-800 DPI

*Additional important information regarding figures and illustrations can be found at <http://bit.ly/EMBOPressFigurePreparationGuideline>

The system will prompt you to fill in your funding and payment information. This will allow Wiley to send you a quote for the article processing charge (APC) in case of acceptance. This quote takes into account any reduction or fee waivers that you may be eligible for. Authors do not need to pay any fees before their manuscript is accepted and transferred to our publisher.

***** Reviewer's comments *****

Referee #2 (Remarks for Author):

The authors have done a extremely thorough job at addressing each Reviewers' comments and questions. They have included new analyses and the revised the manuscript is greatly improved. This paper will be of high interest to the field.

The authors performed the requested editorial changes.

Corresponding Author Name: José Luis Molinuevo, Kaj Blennow

Manuscript Number: EMM-2020-12921